



**Biomass Burning Aerosols and the Low Visibility Events in Southeast Asia**

Hsiang-He Lee[1@], Rotem Z Bar-Or[2], and Chien Wang[1,2]

[1] Center for Environmental Sensing and Modeling, Singapore-MIT Alliance for

Research and Technology, Singapore

[2] Center for Global Change Science, Massachusetts Institute of Technology,

Cambridge, MA, U.S.A.

Submitted to

Atmospheric Chemistry and Physics

June 14, 2016

@Corresponding author address: Dr. Hsiang-He Lee, 1 Create Way, #09-03 CREATE
Tower, Singapore, 138602
E-mail: hsiang-he@smart.mit.edu



**Abstract**
Fires including peatland burning in Southeast Asia have become a major
concern of general public as well as governments in the region.  This is because that
aerosols emitted from such fires can cause persistent haze events under favorite
weather conditions in downwind locations, degrading visibility and causing human
health issues.  In order to improve our understanding of the spatial-temporal
coverage and influence of biomass burning aerosols in Southeast Asia, we have used
surface visibility and particulate matter concentration observations, added by
decadal long (2002 to 2014) simulations using the Weather Research and
Forecasting (WRF) model with a fire aerosol module, driven by high-resolution
biomass burning emission inventories.  We find that in the past decade, fire aerosols
are responsible for nearly all the events with very low visibility (< 7km), and a
substantial fraction of the low visibility events (visibility < 10 km) in the major
metropolitan areas of Southeast Asia: 38% in Bangkok, 35% in Kuala Lumpur, and
34% in Singapore.  Biomass burnings in Mainland Southeast Asia account for the
largest contributor to total fire produced $PM_{2.5}$ in Bangkok (99.1%), while biomass
burning in Sumatra is the major contributor to fire produced $PM_{2.5}$ in Kuala Lumpur
(49%) and Singapore (41%).  To examine the general situation across the region, we
have further defined and derived a new integrated metric for 50 cities of the
Association of Southeast Asian Nations, i.e., Haze Exposure Days (HEDs) that
measures the annual exposure days of these cities to low visibility (< 10 km) caused
by particulate matter pollution.  It is shown that HEDs have increased steadily in the
past decade across cities with both high and low populations.  Fire events are found



to be responsible for about half of the total HEDs. Therefore, our result suggests
that in order to improve the overall air quality in Southeast Asia, mitigation policies
targeting at both biomass and fossil fuel burning sources need to be put in effect.




## 1 Introduction


In recent decades, biomass burning has become frequent and widely spread
across the mainland of Southeast Asia to the islands of Sumatra and Borneo
(Langner et al., 2007; Carlson et al., 2012; Page et al., 2002; van der Werf et al.,
2010). Abundant particulate matters emitted from such fires cause the haze events
to occur in the downwind locations such as Singapore (Koe et al., 2001; Heil et al.,
2007; See et al., 2006), degrading visibility and threatening on human health
(Emmanuel, 2000; Kunii et al., 2002; Johnston et al., 2012; Mauderly and Chow,
2008). Besides causing air quality issues, fire aerosols contain rich carbonaceous
compounds such as black carbon (BC) (Fujii et al., 2014) and thus can reduce
sunlight through both absorption and scattering. Based on satellite data and
numerical simulations, Tosca et al. (2010) found that tropospheric heating from BC
absorption in the Maritime Continent (MC) is $20.5\pm9.3$ W m$^{-2}$, and the reduction of
both surface net shortwave radiation and regional precipitation can be as high as
10% due to the direct and semi-direct effects of fire aerosols. Nevertheless, indirect
effects of fire aerosols are even more complicated due to various cloud types and
meteorological conditions in the MC (Sekiguchi et al., 2003; Lin et al., 2013; Wu et
al., 2013).
Majority of present day fires in Southeast Asia occurs due to human
interferences: oil palm plantation related land clearing, deforestation, and peatland
management, and burning of agriculture wastes (Dennis et al., 2005; Miriam et al.,
2015b). Certain policies and regulations regarding, e.g., migration also affect the
occurrence of burning events. For example, large fires have occurred since 1960s in





Sumatra; however, the first fire event in Kalimantan happened in the 1980s (Field et
al., 2009). Based on economic incentives and population growth in Southeast Asia,
future land-use management will play an important role in determining the
coverage of fires across the region (Carlson et al., 2012; Miriam et al., 2015a).
Besides human interventions, meteorological factors, such as rainfall, can also
influence fire initiation, intensity, and duration (Reid et al., 2012; Reid et al., 2015).
Reid et al. (2012) investigated relationships between fire hotspot appearance and
various climate variabilities as well as meteorological phenomena in different
temporal scales over the MC, including: (1) El Nino and Southern Oscillation (ENSO)
(Rasmusson and Wallace, 1983) and the Indian Ocean Dipole (IOD)(Saji et al., 1999);
(2) Seasonal migration of the Inter-tropical Convergence Zone (ITCZ) and associated
Southeast Asia monsoons (Chang et al., 2005); (3) Intra-seasonal variabilities such
as Madden-Julian Oscillation (MJO) (Madden and Julian, 1971) and the west
Sumatran low (Wu and Hsu, 2009); (4) Wave, mesoscale features, and tropical
cyclones; and (5) Convections. One interesting finding is that the influence of these
factors on fire events varies over different parts of the MC. For example, the fire
signal in a part of Kalimantan is strongly related to both the monsoons and ENSO. In
contrast, fire activity in Central Sumatra is not as closely tied to the monsoons and
ENSO but MJO signal.
Above climate variabilities or meteorological phenomena affect not only
biomass burning emissions but also fire aerosol transport (Reid et al., 2012).
Seasonal migration of the ITCZ and associated monsoonal circulation dominate
seasonal wind flows, whereas sea breeze, typhoon, or topography determine air





102 flow in smaller spatial scales or shorter temporal scales, all of them play significant

103 roles in determining the transport pathway of fire aerosols (Wang et al., 2013). For

104 example, during the intense haze episode of June 2013, the long lasting situation

105 with "very unhealthy" air pollution level in Singapore was actually caused by an

106 enhanced fire aerosol transport from Sumatra to West Malaysia owing to a tropical

107 storm located in South China Sea. Recently, using a global chemistry transport

108 model combining with a back-trajectory tracer model, Reddington et al. (2014)

109 attempted to attribute particulate pollutions in Singapore over a short time period

110 of 5 years to different burning sites in surrounding regions. The coarse 2.8-degree

111 resolution model used in the study, however, has left many open questions.

112  In this study, we aim to examine and quantify the impact of fire aerosols on the

113 visibility and air quality of Southeast Asia in the past decade. Analyses of

114 observational data and comprehensive regional model simulations have both been

115 performed in order to improve our understanding of this issue. We firstly describe

116 methodologies adopted in the study, followed by the results and findings from our

117 assessment of the fire aerosol on the degradation of visibility in several selected

118 cities and also in the great Southeast Asia. We then discuss the sensitivity of our

119 findings to the use of different meteorological datasets as well as fire emission

120 inventories. The last section summarizes and concludes our work.



## 2  Methodology

### 2.1  The model

In order to address the targeted science question, we have used the Weather Research and Forecasting (WRF) model coupled with chemistry component (WRF-Chem). The WRF model is a compressible, non-hydrostatic regional meteorology model that uses the Arakawa C grid and terrain-following hydrostatic pressure coordinates, and includes various dynamic cores and physical parameterizations for different scientific purposes (Skamarock et al., 2008). The WRF-Chem model is a version of the standard WRF with an additional interactively coupled model of atmospheric chemistry. WRF-Chem simulates atmospheric evolutions of chemical species including particulate matters concurrently with meteorological fields, using the same grid structure, advection scheme, and physics schemes for sub-grid scale transport as in the standard WRF model (Grell et al., 2005). In this study, we use WRF-Chem version 3.6 with a modified chemistry tracer module instead of a full chemistry package. This is for the purpose to focus on the fire aerosol life cycle as the first step, without involving a much more complicated gaseous and aqueous chemical processing calculations. This configuration also lowers the computational burden substantially, and thus enables us to conduct long model integrations to determine the contributions of fire aerosol to the degradation of air quality in the region over the past decade. The numerical simulations are employed within a model domain with a horizontal resolution of 36 km, including 432 × 148 horizontal grid points (Fig. 1), and 31 vertically staggered layers based on a terrain-following pressure coordinate system. The vertical layers are stretched with a higher



resolution near the surface (an average depth of ∼30 m in the first model half layer).
Variables other than vertical velocity and geopotential are stored at the half model
layers.    The time step is 180 seconds.    The physics schemes included in the
simulations are listed in Table 1.    The initial and boundary meteorological
conditions are taken from reanalysis meteorological dataset.  In order to examine
the potential influence of different reanalysis products on simulation results, we
have used two such datasets: (1) the National Center for Environment Prediction
FiNaL (NCEP-FNL) reanalysis data (National Centers for Environmental Prediction,
2000), which has a spatial resolution of 1 degree and a temporal resolution of 6
hours; and (2) ERA-Interim, which is a global atmospheric reanalysis from
European Centre for Medium-Range Weather Forecasts (ECMWF) (European Centre
for Medium-Range Weather, 2009), providing 6-hourly atmospheric fields on sixty
pressure levels from surface to 0.1 hPa with a horizontal resolution of
approximately 80 km.  Sea surface temperature is updated every 6 hours in both
NCEP-FNL and ERA-Interim.    All simulations used four-dimensional data
assimilation (FDDA) to nudge NCEP-FNL or ERA-Interim temperature, water vapor,
and zonal and meridional wind speeds above the planetary boundary layer (PBL).
This approach has shown to provide realistic temperature, moisture, and wind fields
in a long simulation (Stauffer and Seaman, 1994).

In WRF-Chem, the sinks of $PM_{2.5}$ particles include dry deposition and wet

scavenging calculated at every time step.



## 2.2 Biomass burning emissions


Two biomass burning emission inventories are used in this study to investigate
the sensitivity of modeled fire aerosol concentration to different emission
estimations. The first emission inventory is the Fire INventory from NCAR version
1.5 (FINNv1.5) (Wiedinmyer et al., 2011), which classifies burnings of extra tropical
forest, topical forest (including peatland), savanna, and grassland. It is used in this
study to provide daily, 36 km resolution $PM_{2.5}$ emissions. The second emission
inventory is the Global Fire Emission Database with version 4.1 with small fire
included (GFEDv4.1s) (van der Werf et al., 2010; Randerson et al., 2012; Giglio et al.,
2013). GFEDv4.1s provides $PM_{2.5}$ emissions with the same spatiotemporal
resolution as FINNv1.5.
A plume rise algorithm for fire emissions was implemented in WRF-Chem by
Grell et al. (2011) to estimate fire injection height. This algorithm, however, often
derives an injection height for tropical peat fire that is too high comparing to the
estimated value based on remote sensing retrievals (Tosca et al., 2011). Therefore,
we have limited the plume injection height of peat fire within 700 m in this study
based on Tosca et al. (2011). This modification has clearly improved the modeled
surface $PM_{2.5}$ concentration comparing to observations in Singapore.
In order to distinguish the spatial-temporal coverage and influence of biomass
burning aerosols from different regions in Southeast Asia and nearby northern
Australia, we have created five tracers to represent fire aerosols respectively from
Mainland Southeast Asia (s1), Sumatra and Java islands (s2), Borneo (s3), the rest of
the Maritime Continent (s4), and northern Australia (s5) as illustrated in Fig. 1. The



major fire season in Mainland Southeast Asia (s1) is from February to April.  In
other four regions (s2-s5), it is from August to October.

Generally speaking, there are strong seasonal variations of fire emissions

coordinating with those of rainfall in all fire regions as shown in Fig. 2.  Because
Mainland Southeast Asia (s1) and northern Australia (s5) are on the edge of
seasonal migration of the ITCZ, seasonal variations of rainfall in these two regions
are even more pronounced.  Sumatra (s2), Borneo (s3), and the rest of the Maritime
Continent (s4) are all influenced by similar meteorological regimes, i.e., seasonal
migration of the ITCZ.  However, the passage of MJO events adds more intra-season
variability of rainfall and fire emissions in these three regions.  Therefore, the
seasonal variations of rainfall and fire emissions in s2, s3, and s4 are not as apparent
as in the s1 and s5 regions (Fig. 2b – d), owing to the influences of multiple scales of
precipitation features over these areas.  Nevertheless, inter-seasonal variations of
rainfall and fire emissions are still highly correlated with each other in these three
regions (see additional discussion in Section 4).

### 2.3   Observational data and model derivation of visibility

The definition of "visibility" is the farthest distance at which one can see a large,

black object against a bright background at the horizon (Seinfeld and Pandis, 2006).
There are several factors to determine visibility, but in this study we mainly
consider the absorption and scattering of light by gases and particles excluding fog
or misty days.  One of the most widely used equations, *Koschmeider equation*, is
given by
$$VIS = 3.912 / b_{ext}, \tag{1}$$





where *VIS* is visibility with a unit in meter and $b_{ext}$ is the extinction coefficient with a
unit of m$^{-1}$. Visibility degradation is most readily observed from the impact of
particulate pollution besides fog. Based on Eq. (1), a maximum visibility under
absolutely dry and pollution-free air is about 296 km owing to Rayleigh scattering,
while a visibility on the order of 10 km is considered as a moderately to heavily
polluted air by particulate matters. Abnormal and persistent low visibility
situations are also referred to as "haze" events. Urban air pollutions such as fossil
fuel burning can cause low visibility and haze event to occur. Similarly, fire aerosols,
alone or mixed with other particulate pollutants, can degrade visibility and lead to
haze events too.
The observational data of visibility from the Global Surface Summary of the Day
(GSOD) (Smith et al., 2011) are used in our study, as derived from the Integrated
Surface Hourly (ISH) dataset and archived at the National Climatic Data Center
(NCDC). The daily visibility in the dataset is available from 1973 to present.
In order to compare with observations, we also calculate the visibility using
modeled fire aerosol data, based on the extinction coefficient of these aerosols as
functions of particle size (assuming a log-normal size distribution of accumulation
mode, with a standard deviation σ = 2), the complex refractive index of the particles,
and a wavelength of 550 nm of the incident light. As fire plumes contain both sulfur
compounds and carbonaceous aerosols, we assume the fire aerosols are aged
internal mixtures with black carbon as core and sulfate as shell (Kim et al., 2008).
We also consider hydroscopic growth of sulfate fraction of these mixed particles in
the calculation based on environmental relative humidity.



As mentioned above, a visibility of 10 km is considered as under moderately to
heavily particulate pollution so that this quantity is used as the threshold for
deriving the "low visibility day (VLD)" in our study.  In analysis, we derived firstly
the low visibility days in every year for a given city using the GSOD visibility data.
Such day is identified when the daily averaged visibility in the observation site is
lower or equal to 10 km.  Then, we derived the low visibility days in the same
procedure, but using modeled visibility data that were only influenced by fire
aerosols.  Both the observed and modeled visibilities were then used to define the
fraction of low visibility days caused by fire aerosols.  It is assumed that whenever
fire aerosol alone could cause a low visibility day to occur, such a day would be
attributed to fire aerosol caused LVD, regardless whether other coexisting
pollutants would have an intensity to cause low visibility or not.  We have also used
a daily visibility of 7 km as the criterion to define the "very low visibility day
(VLVD)".  Such heavy haze events in the region are generally caused by severe fire
aerosol pollution, thus we use their occurrence specifically to evaluate the model
performance.
**2.4   Numerical simulations**
Our simulations cover a time period slightly longer than a decade from 2002 to
2014 based on availability of biomass burning emission estimations.  The simulation
of each year started on 1 November of the previous year and lasted for 14 months.
The first two months are used for spin-up.
Three sets of decadal long simulations have been conducted. The first
simulation used reanalysis data of NCEP-FNL and fire emission inventory of





FINNv1.5. This simulation is hereafter referred to as FNL_FINN and discussed as the
base simulation. In order to examine the influence of different meteorological
inputs on fire aerosol life cycle, the second simulation was conducted using the
same FINNv1.5 fire emission inventory as in FNL_FINN but a different reanalysis
data of ERA_Interim, referring to as ERA_FINN. In addition, to investigate the
variability of fire aerosol concentration brought by the use of different estimations
of fire emissions, the third simulation, FNL_GFED, was driven by the same NCEP-
FNL meteorological input as in FNL_FINN but a different fire emission inventory, the
widely used GFEDv4.1s. Since the daily emission of GFEDv4.1s is only available
after 2003, the period of the FNL_GFED simulation is from 2003 to 2014.
Precipitation is one of the key factors in determining the transport and
scavenging of fire aerosols. WRF simulation driven by NCAR_FNL reanalysis data, or
the FNL_FINN run, produced a monthly mean precipitation of 6.81±0.55 mm day$^{-1}$
over the modeled domain for the period from 2002 to 2014, very close to the value
of 6.29±0.43 mm day$^{-1}$ produced in another simulation driven by ERA_Interim, or
the ERA_FINN run. Comparing to the monthly mean of 4.69±0.38 mm day$^{-1}$ from the
satellite retrieved precipitation in the Tropical Rainfall Measuring Mission (TRMM)
3B43 (V7) dataset (Huffman et al., 2007), however, both results appear to be higher.
Based on the sensitivity tests for FDDA grid nudging, the wet bias in both
experiments mainly comes from water vapor nudging. Figure 3a – c are the
Hovmöller plot of daily TRMM, FNL_FINN, and ERA_FINN precipitation in 2006,
respectively. Comparing to the observations, both FNL_FINN and ERA_FINN have
produced more light rain events, and this appears to be the reason behind the model





precipitation bias. Despite the model overestimation in averaged total precipitation,
the temporal correlation of normalized rainfall anomaly between FNL_FINN
(ERA_FINN) and TRMM is 0.69 (0.90) and the spatial correlation is 0.86 (0.85)
during 2002-2014. The comparisons show that simulated rainfall generally agrees
with the observation in space and time, especial when ERA-Interim reanalysis is
used (i.e., in ERA_FINN).
**3    Assessment of the impact of fire aerosols on Southeast Asia visibility**
**3.1    Impact of fire aerosols on the visibility in four selected cities**

We first to focus our analysis on four selected cities in the region, Bangkok

(Thailand), Kuala Lumpur (Malaysia), Singapore (Singapore), and Kuching
(Malaysia), all located close to the major Southeast fire sites ranging from the
mainland to the islands. Specifically, Bangkok is a smoke receptor city of the fire
events in the mainland of Southeast Asia (s1) while Kuala Lumpur and Singapore
are two cities frequently under the influence of Sumatra (s2) as well as Borneo fires
(s3). Kuching is in the coast area of Borneo so that directly affected by Borneo fire
events (s3).

The low visibility events in these four near-fire-site cities during the fire

seasons from 2002 to 2014, defined as days with daily averaged visibility lower or
equal to 10 km, or Low Visibility Days (LVDs), have been identified using the daily
GSOD visibility database and then compared with modeled results (Fig. 4). We find
that the model has reasonably captured the LVDs despite certain biases.
Specifically, for the Very Lower Visibility Days (VLVDs), here defined as events with



daily averaged visibility lower or equal to 7 km, the modeled and observed results
display a good correlation despite a model overestimate in visibility value or
underestimate in degrading visibility in certain events. In Southeast Asia, severe
haze events equivalent to the VLVDs in visibility degradation are largely caused by
fire aerosol pollutions. Assuming this is true, the performance of our model in
reproducing the major fire events is very good since only 10% or fewer VLVDs
observed in the past decade were not captured by the model (Table 2; Fig. 4). Note
that other than these VLVDs, for many LVDs fire aerosol might not be the only
reason responsible for the degradation of visibility.

In addition to the visibility data, we have also obtained the ground-based

observations of $PM_{2.5}$ concentration in recent years from the National Environment
Agency (NEA) of Singapore. Figure 5a shows the comparison of time series of
observed and FNL_FINN simulated daily $PM_{2.5}$ during 2013-2014. Note that the
observed $PM_{2.5}$ level reflects the influences of both fire and non-fire aerosols,
whereas the modeled $PM_{2.5}$ only includes the impact of fire aerosols. However,
model still predicted clearly high $PM_{2.5}$ concentrations during most of the observed
haze events, especially in June 2013 and in spring and fall seasons of 2014
(highlighted green areas), though with underestimates in particle concentration of
up to 30-50%, likely due to the model resolution, a model overestimation of rainfall,
and the errors in emission inventory. Once again, the model has shown a solid
performance in capturing all the major known haze events caused by fire PM in
Singapore (Fig. 5b). Specifically to the observed VLVDs, we evidence that fire
aerosol is the main reason behind these events.





We find that the annual mean LVDs in Bangkok has increased from 46% in the
first 5-year period of the simulation duration (2002-2009) to 74% in the last 5-year
period (2010-2014), so does the LVDs caused by fire aerosols (Fig. 6a).  Overall, fire
aerosols are responsible for more than one third of these LVDs (i.e. 38% in average;
Table 2).  The largest source of fire aerosols affecting Bangkok is agriculture waste
and other biomass burning in s1 during the dry season of spring (Fig. 7a; Table 3).
During the fire season, abundant fire aerosols degrade visibility and even cause
VLVDs to occur (Fig. 6e).  Ninety-eight percent of VLVDs in Bangkok occurred from
December to April.  Based on our model results, 89% of VLVDs can be identified as
fire caused.
In Kuala Lumpur, the percentage of LVDs also gradually increases since 2006 to
reach a peak in 2011 and again in 2014 (Fig. 6b).  During 2005-2010 the frequency
of total LVDs have increased 10-15% each year, mainly attributing to the pollution
sources other than fires.  However, fire-caused LVDs are more evident after 2009.
Seasonal wise, there are two peaks of fire aerosol influence, one in February-March
and another in August (Fig. 6f), corresponding to the trans-boundary transport of
fire aerosols from Mainland Southeast Asia (s1) in the winter monsoon season and
from Sumatra (s2) in the summer monsoon season, respectively (Fig. 7b).  Three
quarter of VLVDs are occurred in the summer monsoon season due to Sumatra fires.
Noted that in November and December the percentage of LVDs is over 50% and
dominated by the pollutants other than fire aerosols.  These non-fire aerosols come
from either local sources or the areas further inland riding on the winter monsoon





circulation.  Overall, fire pollution is responsible for 35% or a substantial fraction of
total low visibility events in Kuala Lumpur during 2002-2014 (Table 2).

The percentage of LVDs in Singapore has been rapidly increasing since 2012

(Fig. 6c).  Except for 2014, this increase is mostly from anthropogenic pollution
other than fires, especially in 2012 and 2013.  High percentage of LVDs in November
and December could be induced by aerosols from further inland of Mainland
Southeast Asia through long-range transport driven by the monsoon circulation
(Fig. 6g).  Similar to Kuala Lumpur, there are two peaks of fire aerosol influence, one
in February-March and another in September-October (Fig. 6g).   The trans-
boundary transported fire aerosols can come from both Sumatra (s2) and Borneo
(s3) in the summer monsoon season (Fig. 7c).  Except for the severe haze events in
June 2013, VLVDs basically occur in September and October (i.e. 92%) due to both
Sumatra and Borneo fires.  In general, 34% of LVDs in Singapore are caused by fire
aerosols in the  FNL_FINN  simulation  and  the  rest  by  local  and  long-range
transported pollutants (Table 2).  Fire aerosol is still the major reason for the
episodic severe haze conditions.

Because of its geographic location, Kuching is affected heavily by local fire

events during the fire season (Fig. 7d).  Fire aerosols can often degrade the visibility
easily to lower than 7 km and even reach 2 km  (Fig. 4d).  The LVDs mainly occur in
August and September during the fire season (Fig. 6d and h).  The frequency of LVDs
in Kuching is similar to Singapore; however, 25% of those LVDs are considered to be
VLVDs in Kuching while only 4% are in Singapore in comparison (Table 2).





**3.2  Impact of fire aerosols on the visibility in the greater Southeast Asia**
Air quality degradation caused by fires apparently occurs in regions beyond the
above-analyzed four cities. To examine such degradation in the greater Southeast
Asia, we have extended our analysis to cover 50 cities of the Association of
Southeast Asian Nations (ASEAN). The impact of particulate pollution on the
greater Southeast Asia is measured by a metric of "Haze Exposure Day" (HED). HED
can be defined in a population weighted format for the 50 analyzed cities, indicating
the relative exposure of the populations in these cities to the low visibility events
caused by particulate pollution, thus calculated as:
$$HED_{pw} = \sum_{i=1}^{N} C_{pw}(i),\tag{2}$$
here,
$$C_{pw}(i) = pop(i) \cdot C(i)/\sum_{i=1}^{N} pop(i),\tag{3}$$
where $N$ equals to the total number of cities, or 50, $i$ is the index for the 50 analyzed
cities, $C_{pw}(i)$ is the population-weighted fraction of the total Haze Exposure Days and
$pop(i)$ is the population for a given city, $C(i)$ represents the annual LVDs for that city
calculated from the GSOD dataset. Note that we assume that the population of each
city is constant throughout the analyzed period. Another assumption of $HED_{pw}$ is
that everyone in a given city would equally expose to the particulate pollution. The
top four among the 50 cities that made the largest contributions to the $HED_{pw}$ are
Jakarta, Bangkok, Hanoi, and Yangon, with population ranking of 1, 2, 4, and 5,
respectively (Fig. 8a).





In addition, HED can be also defined in an arithmetic mean format, assuming
each city weights equally regardless of its population. Its value hence emphasizes
on the relative exposure of each area within the analyzed region:

$HED_{ar} = \sum_{i=1}^{N} C(i)/N,$                     (4)

Apparently, both $HED_{pw}$ and $HED_{ar}$ can be also calculated using fire-caused LVDs
(here using the results of FNL_FINN) to define the absolute and relative
contributions of fire aerosols to the total low visibility events in the region. We will
label the fire-caused HED as $fHED_{pw}$ and $fHED_{ar}$ thereafter.
We find that both $HED_{pw}$ and $HED_{ar}$ increase rather steadily over the past
decade (Fig. 8b), demonstrating that the exposure to haze events either weighted by
population or not has become worse in the region. Generally speaking, the fire
aerosols are responsible for 40-60% of the total exposures to low visibility across
the region. In both measures, the increase of fire-caused HED (2.64 and 3.37 days
per year for population-weighted and arithmetic mean, respectively) is similar to
that of overall HED (2.61 and 3.59 days per year for population-weighted and
arithmetic mean, respectively) (Fig. 8b), suggesting that fire aerosol has taken the
major role in causing the degradation of air quality in Southeast Asia comparing to
the non-fire particulate pollution. The result that $HED_{pw}$ is higher than $HED_{ar}$ in
most of the years indicates that the particulate pollution is on average worse over
more populous cities than the others. Interestingly, the discrepancy of these two
variables, however, has become smaller in recent years and even reversed in 2014,
implying an equally worsening of haze event occurrence across from the smaller to
the bigger cities in terms of population in the region. The reason behind this result





could be a widely spread of fire events in the region, particularly causing acute haze
events in the cities with relatively low populations. Regarding the increase of fire-
caused HED, because biomass burning, especially peatland burning usually occurs in
the rural areas, higher fire emissions would extend low visibility condition to a
larger area regardless of its population. On the other hand, air pollution caused by
industrialization, urbanization, and other factors such as population growth
increases rapidly across the region so that even cities with lower population now
increasingly suffer from low visibility from fossil fuel burning and other sources of
particulate pollution. Therefore, the mitigation of air quality degradation needs to
consider both fire and non-fire sources.
**3.3   The influence of wind and precipitation on fire aerosol life cycle**

Seasonal migrations of the ITCZ and associated summer and winter monsoons

dominate seasonal wind flows that drive fire aerosol transport. Additionally, as
discussed previously, certain small scale or short-term phenomena such as sea
breeze, typhoon, and topography forced circulations also play important roles in
distributing fire aerosols. Nevertheless, we focus our discussions here on the
former.

February to April is the main fire season in Mainland Southeast Asia (s1). In the

FNL_FINN simulation, seasonal mean concentration of $PM_{2.5}$ within the planetary
boundary layer (PBL) can exceed 20 $\mu g\ m^{-3}$ in this region. During this fire season,
the most common wind direction is from northeast to southwest across the region
(Fig. 9a). Fire aerosol plumes with concentration higher than 0.1 $\mu g\ m^{-3}$ can
transport with the main wind westward as far as 7000 km from the burning sites.





In contrast, February to April is not the typical burning season in the islands. Low
fire emissions added by a lack of long-range transport of fire aerosols from the
mainland due to the seasonal circulation result in a low $PM_{2.5}$ level over these
regions (Fig. 9b - d).

Wet scavenging is a major factor to determine the lifetime and thus abundance

of suspended fire aerosols in the air. The effect of wet scavenging of fire aerosols is
reflected from the wet scavenging time calculated using the modeled results. The
wet scavenging time is a ratio of aerosol mass concentration and scavenging rate,
the latter is a function of precipitation rate. Thus, short scavenging time often
indicates high scavenging rate except for the sites with extremely low aerosol
concentration. During February-April, at the ITCZ's furthest southern extent, the
short scavenging time < 1 day around 10°S shows a quick removal of fire aerosols by
heavy precipitation that has prevented the southward transport of aerosols (Fig. 9f).
Whereas, the long scavenging time (> 5 days) in the Western Pacific warm pool,
South China Sea, the Indochina peninsula, Bay of Bengal, and Arabian Sea leads to a
long suspending time of aerosols transported to these regions. During the same
season, over the islands of Sumatra and Borneo, the abundance along with the
likelihood of being transported to other places of fire aerosols, either emitted locally
or trans-boundary transported, are greatly limited by the high scavenging rate
(short scavenging time) over this regions (Fig. 9g and h). South China Sea is in a dry
condition during this time period, therefore, fire aerosols from the northern part of
Philippine can be transported to this region and stay longer than 5 days (Fig. 9i).



The months of August to October, when the ITCZ reaches its furthest northern
extent, mark the major fire season of Sumatra, Borneo and some other islands in the
Maritime Continent (Fig. 10b - d).  Australia fires also mainly occur in this season
(Fig. 10e).  Mean wind flows are from southeast to northwest in the Southern
Hemisphere, and turn to the northeast direction once pass the Equator.  Within the
MC the seasonal variation of rainfall is small during this time, heavy precipitation
and thus short scavenging time (< 3 days) mostly exist along the MJO path (Fig. 10f -
i) (Wu and Hsu, 2009).  The high scavenging rate in the regions close to the fire sites
in the islands shortens the transport distance of fire aerosol plumes with $PM_{2.5}$
concentration > 0.1 µg m$^{-3}$ to less than 3000 km (Fig. 10b - d).  Long scavenging time
(> 5 days) primarily exists in Banda Sea and northern Australia due to the ITCZ
location.  Fire aerosols from Java Island (s2) (Fig. 10g), Papua New Guinea (s4) (Fig.
10i), and northern Australia (s5) (Fig. 10j) can thus suspend in the air for a
relatively long time over these regions.
The above-discussed seasonal features of precipitation and aerosol scavenging
strength help us to better understand the variability of haze occurrence and also to
identify the major source regions of fire aerosols influencing selected Southeast
Asian cities (Fig. 7).  For example, the geographic location of Bangkok, which is
inside the s1 emission region, determines that about 99% fire aerosols is from
sources within the region from December to April (Fig. 7a and Table 3).  Fire
aerosols from all the other burning sites stay at very low level even during the
burning seasons there due to circulation and precipitation scavenging.  For Kuala
Lumpur and Singapore, over 90% of total fire aerosols reached both cities come





from Mainland Southeast Asia (s1) in January–April due to the dominant winter
monsoon circulation.  During May-October, however, the major sources of fire
aerosols shift to Sumatra (s2) and Borneo (s3) aiding by northward wind (Fig. 10b
and c). The monthly variations of $PM_{2.5}$ concentration in Kuala Lumpur and
Singapore also have a largely similar pattern (Fig. 8b and d).  The annual mean
contribution of different emission regions in Kuala Lumpur are 43% from Mainland
Southeast Asia (s1), 49% from Sumatra (s2), 4% from Borneo (s3), 3% from the rest
of Maritime Continent (s4), and 0.4% from northern Australia (s5) in FINL_FINN
(Table 3).  Similar to Kuala Lumpur, there are two peak seasons of the monthly low
visibility days contributed by fire aerosols in Singapore (Fig. 6g), well correlated
with modeled high fire $PM_{2.5}$ concentration (Fig. 7c).  The low visibility days in
March and April mainly are caused by fire aerosols from Mainland Southeast Asia
(s1) under southward wind pattern (Fig. 9a), and those in May to October are
affected by Sumatra (s2) first in May to June, and then by both s2 and s3 (Borneo)
during August to October due to north- or northwest-ward monsoonal circulation
(Fig. 10b and c; also Table 3).  Kuching, similar to Bangkok, is strongly affected by
local fire aerosols (s3) during fire season (July – October).  The annual mean
contribution from Borneo (s3) is 85% while only 7% from Mainland Southeast Asia
(s1) and 5% from Sumatra (s2) (Table 3).

Reddington et al. (2014) applied two different models, a 3D global chemical

transport model and a Lagrangian atmospheric transport model to examine the
long-term mean contributions of fire emissions to $PM_{2.5}$ from different regions in
Southeast Asia.  The contribution from Mainland Southeast Asia to the above-



discussed four selected cities was lower than our result during January-May, likely
due to their use of a different emission inventory and the coarse resolution of their
global model. FINNv1.5 dataset used in our study specifically provides higher $PM_{2.5}$
emissions from agriculture fires (the major fire type in Mainland Southeast Asia)
than GFED4.1s does, the latter is an updated version the dataset (GFEDv3) used in
Reddington et al. (2014) (Fig. 2). The detail comparison of FNL_FINN and
FNL_GFED will be discussed in the following section.
**4   Influence of different reanalysis datasets and emission inventories on**
**modeled fire aerosol abundance**
As discussed in the previous section, meteorological conditions, particularly
wind field and precipitation, could substantially influence the life cycle and
transport path of fire aerosols during the fire reasons; therefore, it is necessary to
examine any potential discrepancies in modeled particulate matter abundance
attributing to the use of different meteorological datasets.
In comparing the two of our simulations, one was driven by the NCAR_FNL (i.e.,
FNL_FINN), another by the ERA_Interim (i.e., ERA_FINN) meteorological input, we
find that the ERA_FINN run consistently produces less precipitation than FNL_FINN
run during the raining seasons over past decade (Fig. 2) (also see the comparison
results of both runs with observations in Section 2.4). Regarding fire aerosol life
cycle, less rainfall in ERA_FINN results in a weaker wet scavenging condition and
thus higher abundance of fire aerosol concentration than in FNL_FINN. We find that
annual mean concentration of fire $PM_{2.5}$ produced in the ERA_FINN run in Bangkok,



Kuala Lumpur, Singapore, and Kuching is 8.8, 5.4, 3.4, and 7.9 µg m$^{-3}$, respectively,
clearly higher than the corresponding results of the FNL_FINN run of 8.0, 4.9, 3.0,
and 7.1 µg m$^{-3}$ (Table 3). In Mainland Southeast Asia, a twenty-one percent lower
rainfall in ERA_FINN causes the significantly different PM$_{2.5}$ concentration
comparing to FNL_FINN result in the fire season (February – April) (Fig. 2a and
11a). In Kuala Lumpur, the difference in fire PM$_{2.5}$ concentration between these two
runs mainly comes from Sumatra (s2) during June to September; however, in
Singapore and Kuching the concentration difference comes from both Sumatra (s2)
and Borneo (s3) in August to October (Fig. 11b - d and Table 3), all corresponding to
the discrepancy of rainfall between FNL_FINN and ERA_FINN in these regions (Fig.
2b and c).

The difference in aerosol scavenging between ERA_FINN and FNL_FINN extends

to a difference as high as 7% and 12% in the resulted LVDs of Bangkok and Kuching,
respectively, (Table 2), though its influence on the results of Kuala Lumpur and
Singapore is much smaller (3~4%). In general, fire PM$_{2.5}$ concentration in
ERA_FINN is about 10% higher than in FNL_FINN; however, the substantial impact
of fire aerosols on LVDs is more sensitive in places near the burning areas, i.e.,
Bangkok and Kuching. Interestingly, a mild increase of VLVDs in the ERA_FINN run
in Bangkok and Kuching (~1%) (Table 2) implies that the occurrence of severe haze
events is less affected by the rainfall difference in the burning areas.

In addition to meteorological inputs, differences various fire emission

estimations could also affect the modeled results. To examine such an influence, we
have compared two simulations with the same meteorological input but different



549 fire emission inventories, the FNL_FINN using FINNv1.5 and FNL_GFED using

550 GFEDv4.1s. The main differences between the two emission inventories appear

551 mostly in Mainland Southeast Asia (s1) and northern Australia (s5) (Fig. 2a and e;

552 Fig. 12a and e). For instance, the peak month of fire $PM_{2.5}$ concentration in Bangkok

553 shifts from March in FNL_FINN to January in FNL_GFED (Fig. 11a), owing to the

554 difference in temporal pattern between the two fire emission inventories (Fig. 2a).

555 Comparing to FINNv1.5, fire emissions in GFEDv4.1s over Mainland Southeast Asia

556 are more than 66% lower (Fig. 2a), and this results in a 40% lower fire $PM_{2.5}$ in

557 Bangkok (Fig. 11a and Table 3). The lower fire $PM_{2.5}$ concentration in FNL_GFED

558 actually produced a visibility that matches better with observation in Bangkok

559 comparing to the result of FNL_FINN (Fig. S1a).

560  The difference in monthly fire emissions over the islands between the two

561 emission inventories is small, with the fire emission in FINNv1.5 generally higher

562 than that in GFEDv4.1s (Fig. 2b – d). However, fire emissions in GFEDv4.1s are

563 much higher during the fire season in the dry years (i.e. 2004, 2006 and 2009) over

564 s2 and s3 (Fig. 12b and c), leading to a modeled mean $PM_{2.5}$ concentration by

565 FNL_GFED in Kuala Lumpur and Singapore that is higher than that by FNL_FINN

566 during the fire season (Fig. 11b and c). On the other hand, the higher $PM_{2.5}$

567 concentration simulated in FNL_GFED during the June 2013 severe haze event in

568 Kuala Lumpur and Singapore is due to the spatiotemporal distribution of fire spots

569 rather than absolute fire aerosol emissions. Based on our simulations, fire aerosols

570 from Sumatra (s2) are mainly responsible for the severe haze event in June 2013

571 (Fig. 7b – c and Fig. S2b – c). During this event, the total amount of fire emissions in



Sumatra (s2) is lower in GFEDv4.1s than FINNv1.5, however, distributed rather
more densely over a smaller area (Fig. 13c and d). As a result, under the same
meteorological condition, the simulated $PM_{2.5}$ in the FNL_GFED simulation reaches
Singapore in a higher concentration that also matches better with observation than
the result of FNL_FINN (Fig. 13b). A similar result also appears in Kuching, where
the difference in modeled $PM_{2.5}$ concentration between the two model runs is likely
related to the difference in spatial or temporal distributions rather than the mean
quantities of $PM_{2.5}$ emissions since the latter are almost the same in both fire
emissions inventories.
The most evident difference between the two emission inventories occurs in
northern Australia, where FINNv1.5 suggests an almost negligible fire aerosol
emission comparing to GFEDv4.1s (Fig. 2e). Therefore, in the FNL_GFED simulation,
Australia fire aerosols play an important role in Singapore air quality, contributing
to about 22% modeled $PM_{2.5}$ concentration in Singapore. In contrast, Australia fires
have nearly no effect on Singapore air quality in the FNL_FINN run (Table 3). Our
results raise the important issue of the sensitivity of modeled aerosol concentration
in downwind areas to the spatiotemporal distribution, besides the absolute
emission amount from the fire spots. A further study regarding this topic would be
much needed.
**5   Summary and Conclusions**
We have examined the extent of the biomass burning aerosol's impact on the air
quality of Southeast Asia in the past decade using visibility and surface $PM_{2.5}$





measurements along with the WRF model with a modified fire tracer module. The
model has shown a good performance in capturing 90% of the observed severe haze
events (visibility < 7 km) occurred in past decade in several cities close to the major
burning sites. Such events are known to be induced mainly by biomass burning. On
the more general cases of particulate pollution, our study suggests that fire aerosols
are responsible for a substantial fraction of the low visibility days (visibility < 10
km) in several cities: 38% in Bangkok, 35% in Kuala Lumpur, 34% in Singapore, and
32% in Kuching.

The life cycle and transport path, and thus spatial and temporal distributions of

fire aerosols are all influenced by meteorological conditions, especially the seasonal
precipitation distribution and atmospheric circulations. These impacts are well
reflected from the variations of abundance of fire aerosols in the selected cities in
analysis. In general, Mainland Southeast Asia is the major contributor during the
Northeast or winter monsoon season in Southeast Asia. In the Southwest or
summer monsoon season, most fire aerosols come from Sumatra and Borneo.
Specifically, fires in Mainland Southeast Asia are accounted for the largest
percentage of the total fire $PM_{2.5}$ in Bangkok (99.2%), and fires from Sumatra are the
major contributor in Kuala Lumpur (51%) and Singapore (42%). Kuching receives
88% of fire aerosols from local Borneo fires.

By comparing the results from two modeled runs with the same fire emissions

but driven by different meteorological inputs, we have examined the potential
sensitivity of modeled results to meteorological datasets. The discrepancy in
modeling the low visibility events due to different meteorological datasets is clearly



evident, especially in the results of Bangkok and Kuching. However, using different
meteorological input datasets does not appear to have influenced the modeled very
low visibility events, or the severe haze events in the cities close to burning sites.

We have also examined the sensitivity of modeled results to the use of different

emission inventories. We find that significant discrepancies of fire emissions in
Mainland Southeast Asia and northern Australia between two emission inventories
used in the study have caused significant difference in modeled fire aerosol
concentration and visibility, particularly in Bangkok and Singapore. For instance,
the contribution to fire aerosol in Singapore from northern Australia changes from
nearly zero in the simulation driven by FINNv1.5 to about 22% in another
simulation driven by GFEDv4.1s. We have also identified the influence of the
discrepancy in spatiotemporal distribution rather than total emitted quantities from
the fire hotspots on modeled $PM_{2.5}$ concentration. Further analysis on this direction
is much needed.

To further assess the impacts of fire events on the air quality of the great

Southeast Asia, we have defined and derived a metric of "Haze Exposure Days"
(HEDs), by integrating annual low visibility days of 50 cities of the Association of
Southeast Asian Nations and weighted by population or averaged arithmetically.
We find that a very large population of Southeast Asia has been exposed to relatively
persistent hazy condition. The top four cities in the HED ranking, Jakarta, Bangkok,
Hanoi, and Yangon, with a total population exceeding two millions, have
experienced more than 200 days per year of low visibility due to particulate
pollution over the past decade. Even worse is that the number of annual low





visibility days have been increasing steadily not only in high population cities but
also those with relatively low populations, suggesting a widely spread of particulate
pollutions into the great Southeast Asian region. Generally speaking, the fire
aerosols are found to be responsible for about half of the total exposes to low
visibility across the region. Our result suggests that in order to improve the air
quality in Southeast Asia, besides reducing or even prohibiting planned or
unplanned fires, mitigation policies targeting at pollution sources other than fires
need to be put in effect as well.

**Acknowledgements.**
This research was supported by the National Research Foundation Singapore
through the Singapore-MIT Alliance for Research and Technology, the
interdisciplinary research program of Center for Environmental Sensing and
Modeling. It was also supported by the U.S. National Science Foundation (AGS-
1339264), U.S. DOE (DE-FG02-94ER61937) and U.S. EPA (XA-83600001-1). The
authors would like to acknowledge the National Environment Agency (NEA) of
Singapore for making Singapore $PM_{2.5}$ data available, the NCEP-FNL, ECMWF ERA-
Interim, NCAR FINN, and GFED working groups for releasing their data to the
research communities, and the NCAR WRF developing team for providing the
numerical model for this study. We thank the National Supercomputing Centre of
Singapore (NSCC) for providing computing resources and technical support.

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




Table 1. WRF physics scheme configuration

| Physics Processes | Scheme |
|---|---|
| microphysics | Morrison (2 moments) scheme |
| longwave radiation | rrtmg scheme |
| shortwave radiation | rrtmg scheme |
| surface-layer | MYNN surface layer |
| land surface | Unified Noah land-surface model |
| planetary boundary layer | MYNN 2.5 level TKE scheme |
| cumulus parameterization | Grell-Freitas ensemble scheme |







Table 2. Annual mean low visibility days (LVDs) and very low visibility days
(VLVDs) per year, and the percentage contributions along with standard deviations
of fire and non-fire (other) pollutions for total low visibility days in Bangkok, Kuala
Lumpur, Singapore and Kuching during 2002-2014 (FNL_GFED is from 2003 to
2014). Parentheses show the percentage of year.

| **FNL_FINN** | **LVD per year (days)** | **Fire pollution contribution (%)** | **Other pollution contribution (%)** |
|---|---|---|---|
| Bangkok, Thailand | 211±49 (58±14%) | 38±8 | 62±8 |
| Kuala Lumpur, Malaysia | 166±80 (45±22%) | 35±18 | 65±18 |
| Singapore, Singapore | 92±84 (25±23%) | 34±16 | 67±16 |
| Kuching, Malaysia | 95±54 (26±15%) | 32±14 | 68±14 |
| **FNL_FINN** | **VLVD per year (days)** | **Fire pollution contribution (%)** | **Other pollution contribution (%)** |
| Bangkok, Thailand | 17±10 (5±3%) | 89±19 | 11±19 |
| Kuala Lumpur, Malaysia | 18±18 (5±5%) | 85±17 | 15±17 |
| Singapore, Singapore | 4±4 (1±1%) | 92±32 | 8±32 |
| Kuching, Malaysia | 24±19 (7±5%) | 94±12 | 6±12 |
| **ERA_FINN** | **VLD per year (days)** | **Fire pollution contribution (%)** | **Other pollution contribution (%)** |
| Bangkok, Thailand | 211±49 (58±14%) | 45±8 | 55±8 |
| Kuala Lumpur, Malaysia | 166±80 (45±22%) | 39±16 | 61±16 |
| Singapore, Singapore | 92±84 (25±23%) | 37±18 | 63±18 |
| Kuching, Malaysia | 95±54 (26±15%) | 44±17 | 56±17 |
| **ERA_FINN** | **VLVD per year (days)** | **Fire pollution contribution (%)** | **Other pollution contribution (%)** |
| Bangkok, Thailand | 17±10 (5±3%) | 90±20 | 10±20 |
| Kuala Lumpur, Malaysia | 18±18 (5±5%) | 90±18 | 10±18 |
| Singapore, Singapore | 4±4 (1±1%) | 98±5 | 2±5 |
| Kuching, Malaysia | 24±19 (7±5%) | 95±11 | 5±11 |
| **FNL_GFED** | **VLD per year (days)** | **Fire pollution contribution (%)** | **Other pollution contribution (%)** |
| Bangkok, Thailand | 215±50 (59±14%) | 36±8 | 64±8 |
| Kuala Lumpur, Malaysia | 174±78 (48±21%) | 28±17 | 72±17 |
| Singapore, Singapore | 96±87 (26±24%) | 29±21 | 71±21 |
| Kuching, Malaysia | 95±57 (26±15%) | 26±18 | 74±18 |
| **FNL_GFED** | **VLVD per year (days)** | **Fire pollution contribution (%)** | **Other pollution contribution (%)** |
| Bangkok, Thailand | 15±8 (4±2%) | 90±19 | 10±19 |
| Kuala Lumpur, Malaysia | 18±18 (5±5%) | 83±28 | 17±28 |
| Singapore, Singapore | 4±4 (1±1%) | 89±37 | 11±37 |
| Kuching, Malaysia | 22±18 (6±5%) | 89±28 | 11±28 |






Table 3. Annual mean and standard deviation of fire PM$_{2.5}$ concentration (µg m$^{-3}$)
contributed by each source region in Bangkok, Kuala Lumpur, Singapore, and
Kuching during 2002-2014 (FNL_GFED is from 2003 to 2014). Parentheses show
the fire aerosol fraction in total PM$_{2.5}$.

| FNL_FINN | s1 | s2 | s3 | s4 | s5 |
|---|---|---|---|---|---|
| Bangkok | 8.0±2.6 (99.1±0.5%) | 0.0±0.0 (0.1±0.1%) | 0.0±0.0 (0.1±0.1%) | 0.0±0.0 (0.7±0.5%) | 0.0±0.0 (0.0±0.0%) |
| Kuala Lumpur | 2.1±1.2 (43.3±14.6%) | 2.5±1.4 (49.3±14.3%) | 0.2±0.1 (4.1±4.4%) | 0.1±0.1 (2.9±2.6%) | 0.0±0.0 (0.4±0.2%) |
| Singapore | 1.0±0.7 (34.3±16.4%) | 1.2±0.8 (40.7±15.3%) | 0.5±0.4 (16.0±11.3%) | 0.2±0.1 (6.7±4.2%) | 0.1±0.0 (2.2±1.1%) |
| Kuching | 0.4±0.4 (7.3±6.6%) | 0.3±0.1 (4.6±2.4%) | 6.3±3.2 (85.3±9.6%) | 0.1±0.1 (2.3±2.3%) | 0.0±0.0 (0.6±0.2%) |
| ERA_FINN | s1 | s2 | s3 | s4 | s5 |
| Bangkok | 8.7±2.7 (99.1±0.4%) | 0.0±0.0 (0.1±0.1%) | 0.0±0.0 (0.1±0.1%) | 0.1±0.0 (0.7±0.4%) | 0.0±0.0 (0.0±0.0%) |
| Kuala Lumpur | 2.1±1.2 (38.6±12.7%) | 3.0±1.5 (53.7±11.9%) | 0.2±0.2 (4.7±4.2%) | 0.1±0.0 (2.6±2.1%) | 0.0±0.0 (0.4±0.2%) |
| Singapore | 1.0±0.6 (31.9±15.3%) | 1.4±0.9 (40.4±13.1%) | 0.7±0.6 (18.9±12.8%) | 0.2±0.1 (6.8±3.7%) | 0.1±0.0 (1.9±1.0%) |
| Kuching | 0.5±0.4 (7.5±5.7%) | 0.4±0.2 (5.9±3.9%) | 6.9±3.8 (83.4±10.1%) | 0.1±0.1 (2.7±2.9%) | 0.0±0.0 (0.6±0.2%) |
| FNL_GFED | s1 | s2 | s3 | s4 | s5 |
| Bangkok | 4.8±1.3 (99.6±0.2%) | 0.0±0.0 (0.1±0.0%) | 0.0±0.0 (0.1±0.1%) | 0.0±0.0 (0.2±0.2%) | 0.0±0.0 (0.1±0.0%) |
| Kuala Lumpur | 1.3±0.6 (38.6±20.8%) | 2.7±1.9 (53.8±21.1%) | 0.1±0.2 (2.8±3.5%) | 0.0±0.0 (0.8±0.8%) | 0.1±0.1 (3.9±3.4%) |
| Singapore | 0.3±0.2 (22.1±17.3%) | 1.5±1.8 (40.2±23.6%) | 0.4±0.5 (12.5±9.5%) | 0.1±0.0 (2.9±2.4%) | 0.4±0.2 (22.3±13.2%) |
| Kuching | 0.1±0.1 (7.2±6.8%) | 0.1±0.1 (4.3±3.2%) | 3.2±3.2 (75.2±12.9%) | 0.0±0.0 (1.7±2.7%) | 0.3±0.2 (11.6±6.7%) |







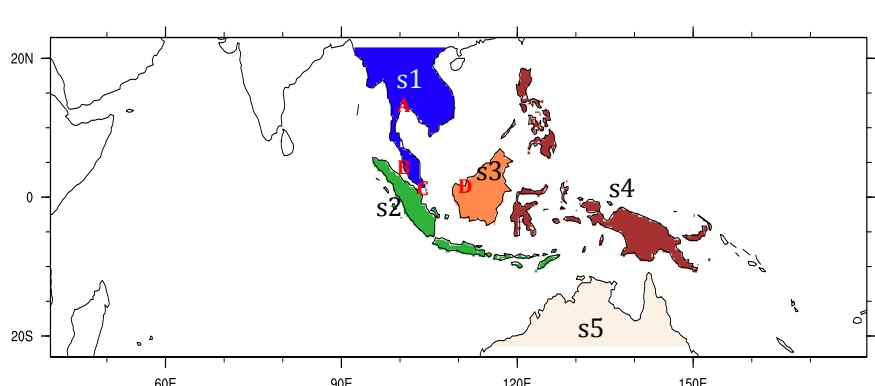

Figure 1. Model domain used for simulations. Domain consists of 31 vertical levels,
each with 432 × 148 grid points with a horizontal resolution of 36 km. Five colored
fire source regions, labeled as s1, s2, s3, s4 and s5, represent Mainland Southeast
Asia, Sumatra and Java islands, Borneo, the rest of Maritime Continent, and northern
Australia, respectively. A, B, C and D indicate the location of four selected cities:
Bangkok, Kuala Lumpur, Singapore and Kuching, respectively.














Figure 2. Monthly $PM_{2.5}$ emissions (Tg year$^{-1}$) in FINNv1.5 (red lines) and GFEDv4.1s
(pink lines). Also shown are precipitation rate (mm day$^{-1}$) simulated in FNL_FINN
(light blue lines) and ERA_FINN (blue lines). All data are averaged during 2002-
2014 for: (a) Mainland Southeast Asia (s1), (b) Sumatra and Java islands (s2), (c)
Borneo (s3), (d) the rest of the Maritime Continent (s4), and (e) northern Australia
(s5). Note that GFEDv4.1s $PM_{2.5}$ emission is averaged from 2003 to 2014.








Figure 3.  Hovmöller (time vs. longitude) plot of daily precipitation in 2006 derived
from: (a) TRMM, (b) FNL_FINN, and (c) ERA_FINN. Latitude average is from 10°S to
10°N. Unit is mm day$^{-1}$.




Figure 4. Comparison of daily visibility between GSOD observation (black lines) and
FNL_FINN modeled result (red lines) in: (a) Bangkok, (b) Kuala Lumpur, (c)
Singapore, (d) Kuching during the fire seasons from 2002 to 2014. Two grey lines
mark the visibility of 7 and 10 km, respectively.




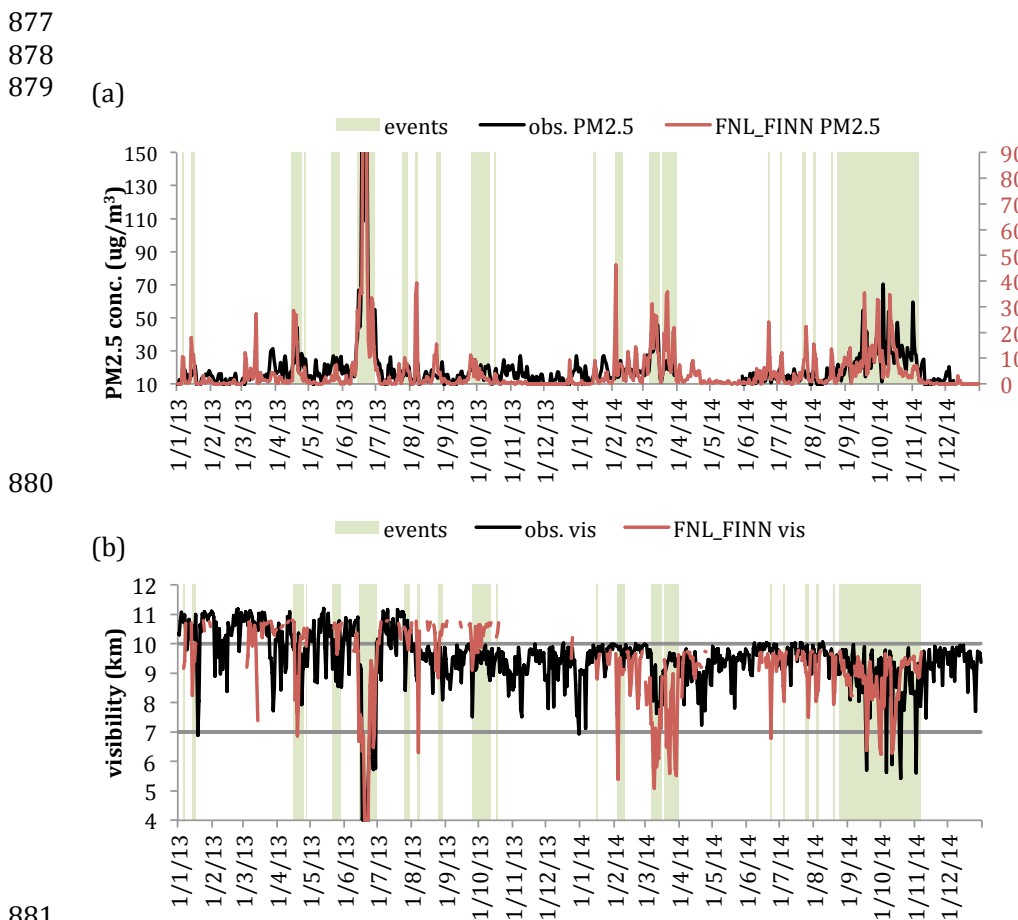



Figure 5. (a) Time series of daily surface $PM_{2.5}$ from the ground-based observations
(black line) and FNL_FINN simulated results (red line) in Singapore during 2013-
2014. (b) Time series of daily visibility of GSOD observation (black line) and
calculated result from FNL_FINN (red line) in Singapore during 2013-2014.
Highlighted green areas are known haze events caused by fire aerosols. Two gray
lines mark the visibility of 7 and 10 km, respectively.




Figure 6. (a) – (d) The annual variation of the percentage of LVDs per year from
GSOD observational visibility in Bangkok, Kuala Lumpur, Singapore, and Kuching,
respectively. (e) – (h) The monthly variation of the percentage of LVDs from GSOD
observational visibility in Bangkok, Kuala Lumpur, Singapore, and Kuching,
respectively, averaged over 2002-2014.

(a)
(b)
(c)
(d)
Figure 7. The monthly variation of mean PM$_{2.5}$ concentration from each emission
regions (s1 - s5) in (a) Bangkok, (b) Kuala Lumpur, (c) Singapore and (d) Kuching,
derived from FNL_FINN simulation and averaged over the period 2002-2014.




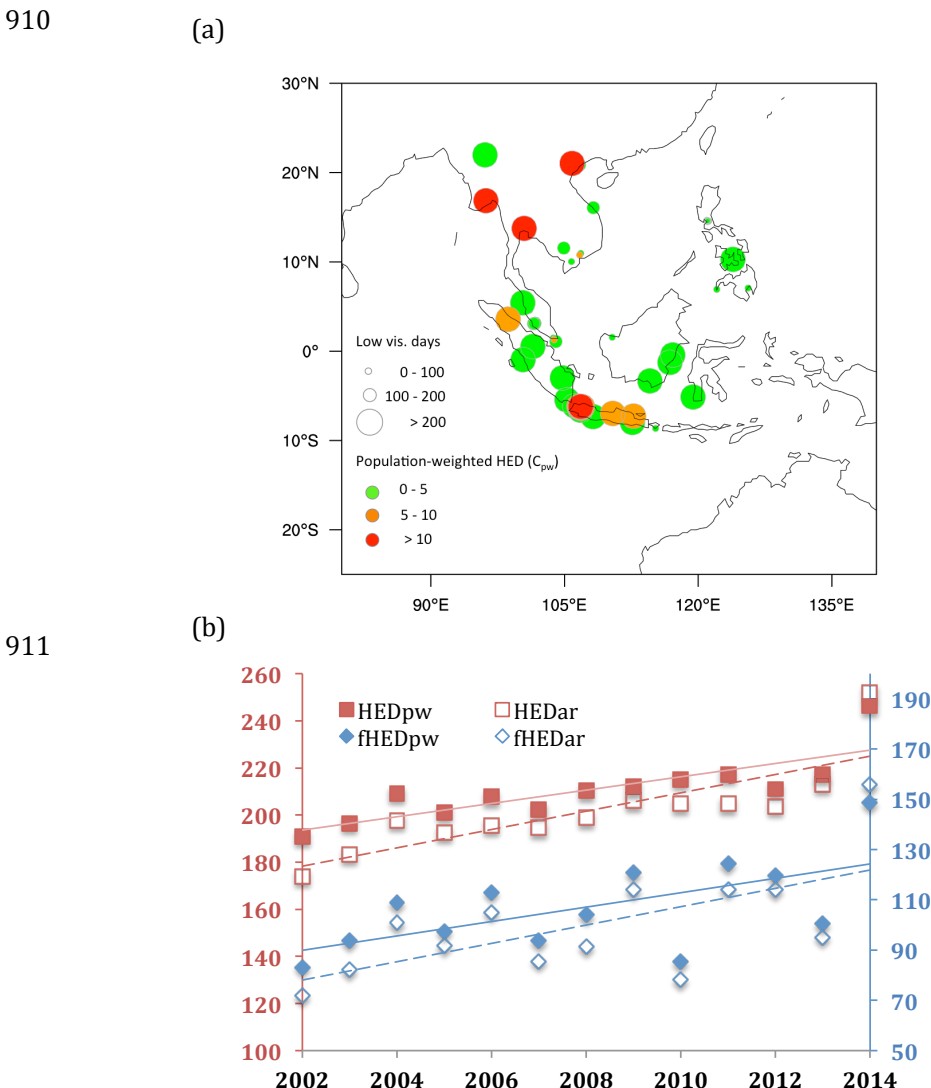


Figure 8. (a) The mean low visibility days (circles) per year from 2002 to 2014 in 50
ASEAN cities and their population-weighted fraction in the total Haze Exposure
Days (HED; colors). (b) Annual variation of population-weighted HED (HED$_{pw}$) and
arithmetic mean HED (HED$_{ar}$). Fire-caused HED are labeled as fHED$_{pw}$ and fHED$_{ar}$.
Units are in days.





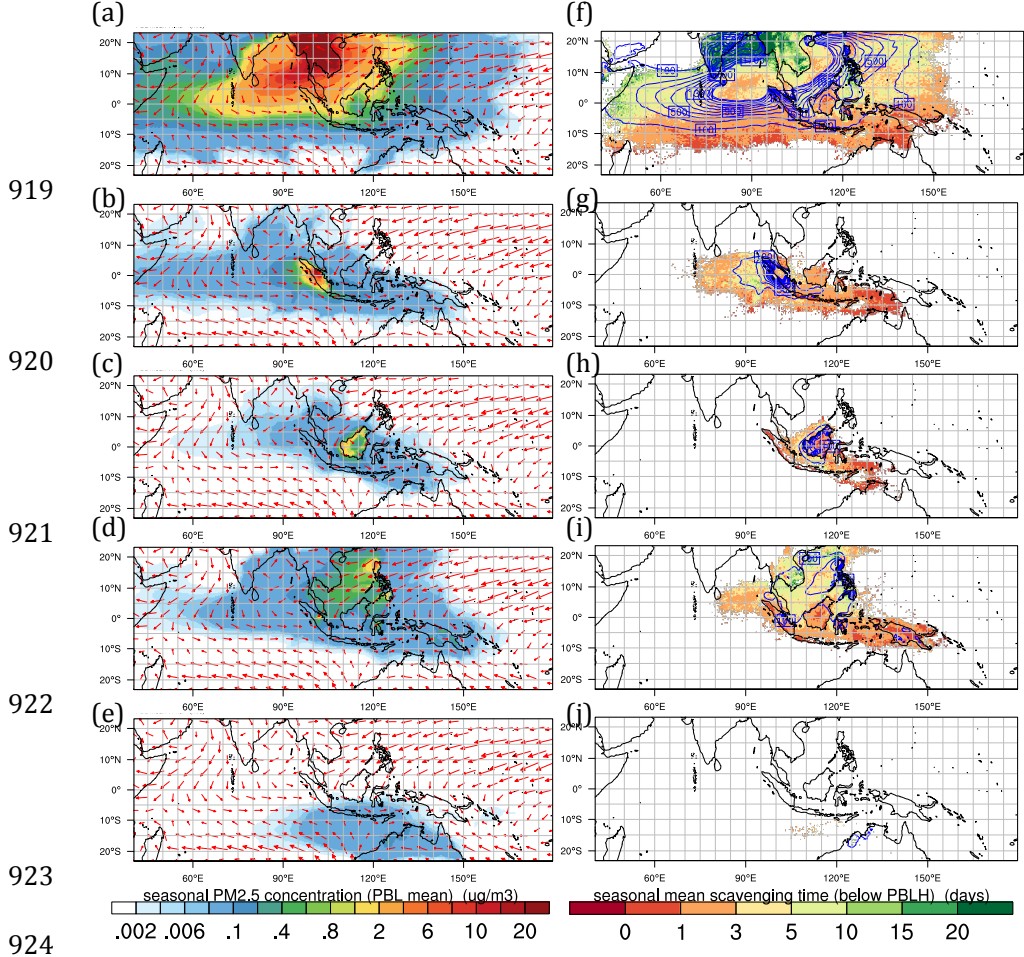

Figure 9. (a)-(e) Seasonal mean PM$_{2.5}$ concentration (µg m$^{-3}$) and wind within the PBL modeled in FNL_FINN during February to April, 2002 – 2014 in: Mainland Southeast Asia (s1), Sumatra and Java island (s2), Borneo (s3), the rest of the Maritime Continent (s4), and northern Australia (s5), respectively. (f)-(g) Same as (a)-(e) but for seasonal mean wet scavenging time (days; shaded) and column intergraded PM$_{2.5}$ concentration (µg m$^{-2}$; contours) within the PBL height.



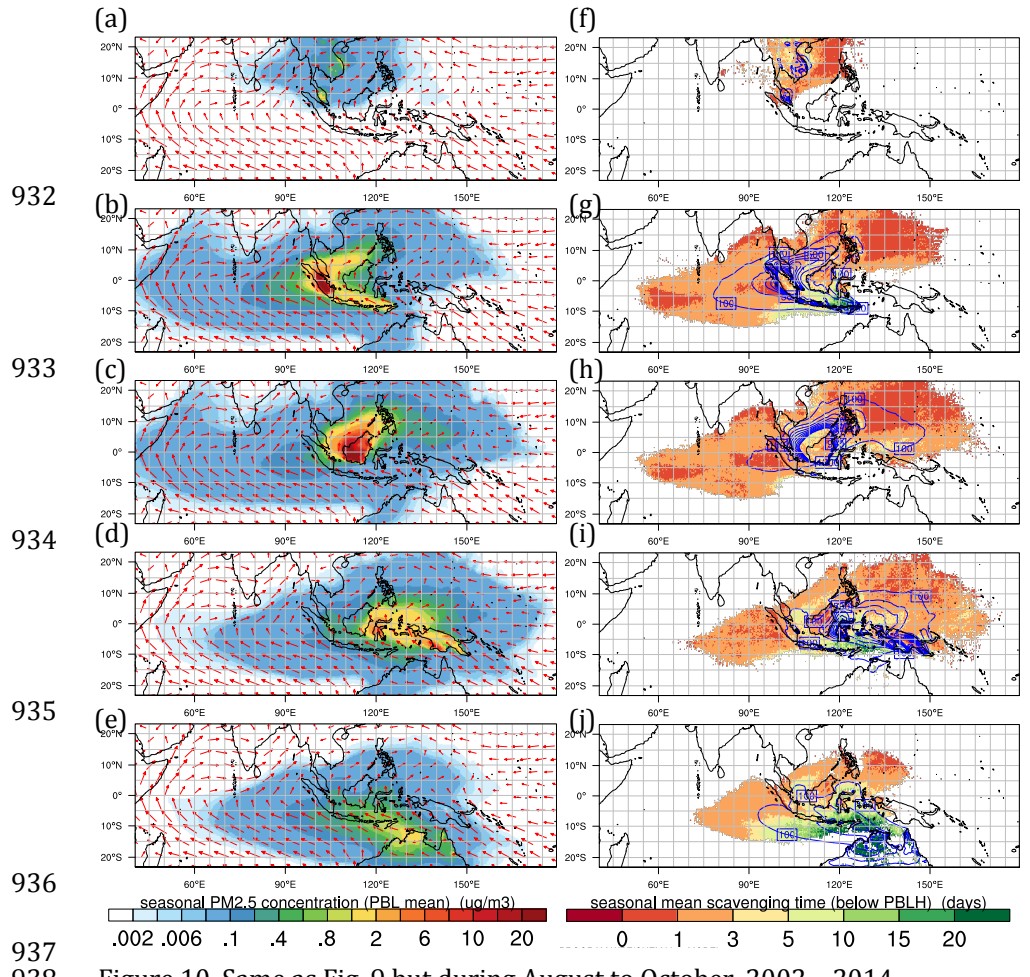

Figure 10. Same as Fig. 9 but during August to October, 2002 – 2014.





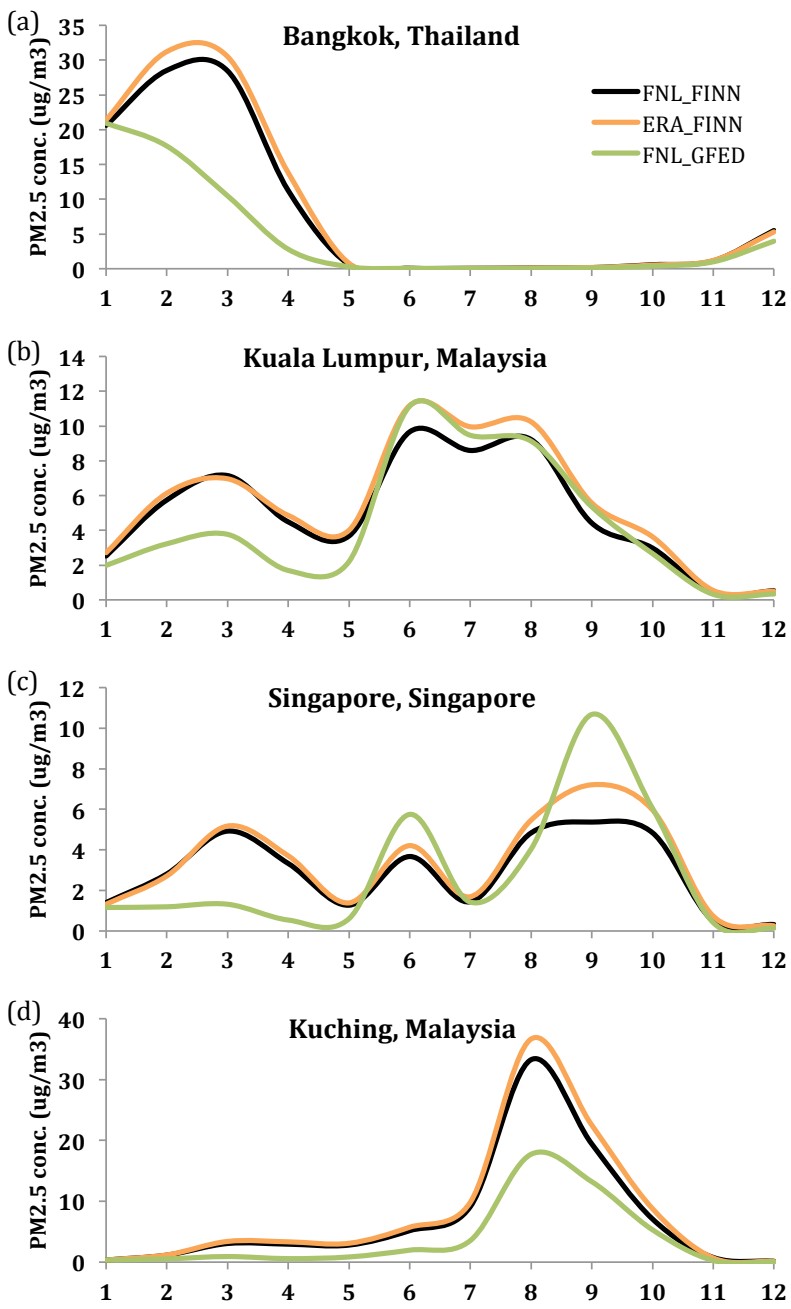

Figure 11. The monthly variation of mean PM$_{2.5}$ concentration in FNL_FINN, ERA_FINN, and FNL_GFED in: (a) Bangkok, (b) Kuala Lumpur, (c) Singapore, and (d) Kuching over the period 2002-2014 (FNL_GFED is from 2003 to 2014).







Figure 12. Temporal variation of monthly $PM_{2.5}$ emission (Tg year$^{-1}$) in FINNv1.5 (pink solid lines) and GFEDv4.1s (red dashed lines). Also shown are precipitation rates (mm day$^{-1}$) simulated in FNL_FINN (light blue solid lines) and ERA_FINN (blue dashed lines) during 2002-2014 in: (a) Mainland Southeast Asia (s1), (b) Sumatra (s2), (c) Borneo (s3), (d) the rest of the Maritime Continent (s4), and (e) northern Australia (s5).





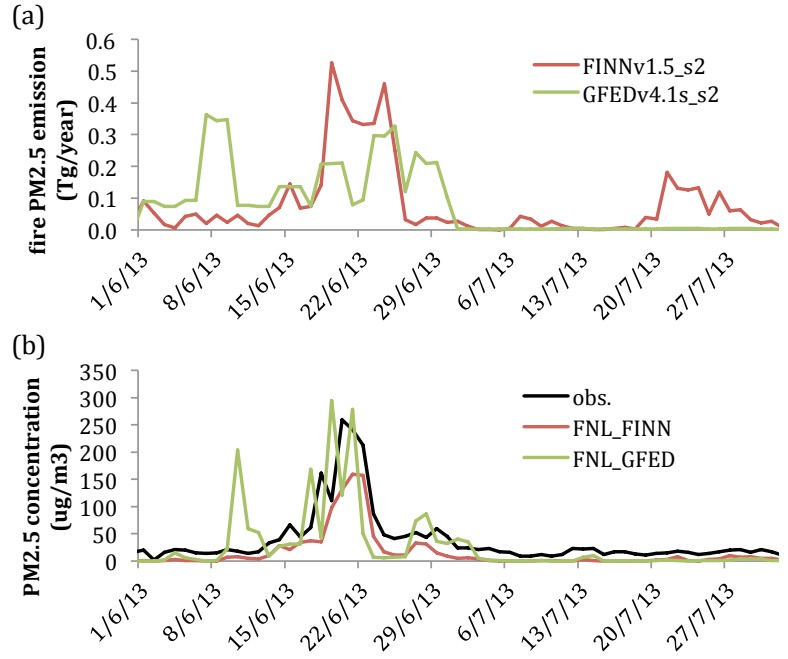

Figure 13. (a) Time series of daily mean $PM_{2.5}$ emissions (Tg year$^{-1}$) in Sumatra (s2) from FINNv1.5 (red line) and GFEDv4.1s (green line). (b) Time series of daily mean $PM_{2.5}$ concentration (µg m$^{-3}$) in Singapore from observation (black line), and modeled results from FNL_FINN (red line) and FNL_GFED (green line). (c) Monthly mean $PM_{2.5}$ emissions (Tg year$^{-1}$) from FINNv1.5 in June 2013. (d) same as (c) but from GFEDv4.1s.