# Peer review of "Biomass Burning Aerosols and the Low Visibility Events in Southeast Asia"

_Atmospheric Chemistry and Physics, 2016_

## Referee Comment (RC1) · Anonymous Referee #1 · 4 Jul 2016

**Review of manuscript doi:10.5194/acp-2016-504, 2016**

**Biomass Burning Aerosols and the Low Visibility Events in Southeast Asia**

**by**

**H. Lee, R.Z. Bar-Or and C. Wang**

**General comments:**

The manuscript addresses an emerging issue for Southeast Asia which concerns the impact of biomass burning on air quality and visibility. The topic is highly relevant for publication in Atmospheric Chemistry and Physics, however major issues related to the form in which the work is structured and presented (i.e. a whole rewriting of the paper is needed), clarifications in methods and analyses need to be addressed. The overall work needs to be synthesized both in the text and in the selection of the figures presented (of the 13 figures included some of them duplicate information included in other ones. If the authors want to keep all of them, they should consider moving some of the figures to the Supplementary Materials).

**Specific comments:**

**Language**

A major rewriting of the paper is needed. Several sentences are not fluent and a grammar/ punctuation check is needed. Below are some examples:

Line 32: remove "that"

Line 33: favorite should be "favourable"

Line 41 and other parts: please be consistent with the tense you use. ….

Line 55: "put in effect", replace with "implemented"

Line 82: please check your references (e.g. Miriam is the first name)

Line 118: "the great Southeast Asia" should be replaced with something similar to "over the whole Southeast Asia". Please check also elsewhere in the paper.

Line 135: please rephrase

Line 168: "estimations" should be always replaced with "estimates"

Line 172: remove "with"

Line 178: "comparing" should be "compared". Please amend this everywhere in the paper.

Line 190-202: please rephrase and summarize. This paragraph is too repetitive and needs to be more concise.

Line 211: units, please replace also elsewhere

Line 236: "this" is missing

Line 294: "so that" is very often used incorrectly. Please check all the occurrences.

Line 343: "are occurred", should be "occurred"

Line 515: "reasons" should be "seasons". Please check also other typos.

Line 518-519: Please rephrase

Line 571-580: this section needs to be rewritten. Sentences are too long and convoluted and several grammar errors are present.

**Methods:**

All the introduction regarding WRF is not needed since you are using a modified version of WRF-Chem. Also you start introducing the model and have section 2.2. describing the emissions and section 2.4 discussing again the simulations. The whole method section has to be reorganized (e.g. have one section discussing the data, one on the model and one on the methods used). Please be more concise and avoid repeating the same information in different sections.

Line 123: please refer more precisely to your "targeted science questions"

Line 139: you mostly focus on visibility so please also add that.

Line 145: this is redundant information, please remove it.

Line 146: The reported time step is for chemistry or physics?

Line 165: Did you only include fire emissions? Does WRF-Chem use other anthropogenic emissions?

Line 208: this should be rephrased by saying what you used for computing visibility.

Line 213-216: please add a reference and rephrase

Line 222: please be more specific by explaining how you will use the GSOD data and to address which objectives

Line 219: add "by increasing bext"

Line 225: Here you introduce model simulations, but you have a section later discussing that. You should reorganize the methods and be more clear on the objectives you are addressing. "In order to compare with observations", what do you mean? Are you referring to a model evaluation? If so please explain in the relevant section how you will perform it.

Line 227: is there a reference you can quote for these assumptions? Or some local measurements used to estimate those parameters?

Line 225-233: this paragraph should be clarified. It is not clear how you link the discussion on fire emission composition, hygroscopic growth, etc. with your work. If it is for general overview purposes, please add it to the introduction or remove it.

Line 238-239: again this is repetition of definitions already given. Please remove this from here and elsewhere in the manuscript.

Line 268: what is the NCAR_FNL? You have not introduced that before. Please add a reference for all datasets used.

Line 267-272: this paragraph needs to be rewritten. Is there any difference between precipitation simulated with NCAR_FNL and FNL_FINN? Otherwise synthesise this result by comparing the simulations run with FNL and ERA. What does it mean "both results appear to be higher"? Please rephrase.

Line 301: LVDs and VLVDs have already been defined so avoid repetitions.

Line 332: how can you distinguish the evets caused by fires? Is it because your simulations do not include other anthropogenic emissions? Otherwise please explain how you conducted your analyses.

Line 349-362: please rephrase to remove repetitions.

**Results**

Line 374-384: this part should be moved to the methods. You need to define earlier how you will conduct your analyses. Also using LVD in equation 3 might be more appropriate than C(i).

Line 432: here it would be also interesting to compare with the WHO limits (i.e. the limit for annual mean $PM_{2.5}$ is 10 µg m$^{-3}$).

Line 590: Section 4 should be rewritten. The way results are presented is too repetitive and convoluted. It would be also easier for the reader to have some clear sentences summarizing the skills of different models/emissions.

**Figures**

Thirteen figures are really too many especially since most of them have several panels. Please select the most critical ones to summarize your findings and move the others to the supplementary material. Also some figures duplicate content shown in other, so either delete them or move to the supplements.

Figure 1: the number of vertical levels cannot be inferred from the figure, so please remove this part of the sentence from the caption. Also, the letters A-D are not easily readable. Please choose different colors.

Figure 2: $PM_{2.5}$ on the y-axis is not as subscript 2.5. It would be easier for the reader to have the whole name of the regions on top of each panel.

Figure 3: is this the yearly average of the daily means? The units can be put after "precipitation".

Figure 5: From panel (a) it is clear that the model highly underestimates observations and a scaling factor is needed. This has to be commented in the text. Could you also start both the y-axes from 0? A scatter plot might also help in quantifying the underestimation or please provide some more statistics for model evaluation.

Figure 6. What do you mean with "variation"? How did you compute it? Please also report the meaning of the color coding in the caption.

Figure 7: Please define "variation" or rephrase. Please do the same for all other figures presenting that wording.

Figure 8: (a) please rephrase saying that the size of the circles indicates the number of days and the colors refer to specific population weights. (b) Please add units on y-axes and mention in the caption the use of different scales.

Figure 9: region s1-s5 are not reported on the panels, so please remove them from the caption and simplify the caption as well. Also it is not clear why you report the results separately by region instead of on one single figure. Figure 9 is essentially identical to Figure 10 averaging on a different period, so you can have just a four panels figure with on each panel a map showing different seasons and the 5 regions together and two panels with the same for wet scavenging. Otherwise you need to move one of the two figures to the supplements.

Figure 11: this is again a repetition of Figure 7. Either you condense the information in one figure or move some of the material to the supplements. It is very hard to keep in mind so many similar figures and your key message is not delivered effectively.

Figure 12: Why do you have y-axes with negative numbers? You are displaying PM concentrations and precipitation, so your minimum value should be zero. This figure again contains information already presented (Figure 11, 7, 13), so please try and condense the figures or move them to the supplements. The captions of all figures should be also more informative on the message you want to deliver to the reader.

---

## Referee Comment (RC2) · Anonymous Referee #2 · 30 Aug 2016

General Comments:

The paper provides a correlation of modelled particulate matter with low visibility days recorded at observation sites across South East Asia. Information is presented about the most likely source areas for biomass burning pollution for different cities and different seasons.

This is an interesting application of an alternative observation dataset for assessing the impact of biomass burning haze on the region and for validating CTM and dispersion models. However, the significant flaw in the way the results are presented is that the model is assumed to be correct and that all low visibility days that are not modelled are therefore specified to be due to other pollution contributions. The validity of this assumption is not demonstrated. It is quite possible that the model is over-estimating

the biomass contribution at some sites and underestimating it at others. Fig 6 for example would suggest that the model may not be capturing up to 50% of the fire haze days, and Fig 4 would suggest that the model misses 50% of the VLVDs at Singapore. The references in the text to fire and non-fire LVD are therefore misleading. The authors need to reconsider how they interpret this data and present it in the paper.

The paper would benefit from some reorganisation of the sections and a reduction in the number of figures.

Specific Comments:

Following on from the general comments, I am concerned that no real attempt at model validation is made within this paper. An additional source of observed data, e.g. PM10 concentrations, from a minimum of one of the sites (ideally many more) is needed to demonstrate that the WRF-Chem simulations are correctly capturing the fire component. The data shown in Fig 5(a) is misleading due to the use of different scales and a more robust analysis of this data is needed earlier in the paper. In fact this data may reveal useful information about missing "background" PM from the model. There are statements on line 320 that the model is underestimating PM2.5 concentration by up to 30-50% in this comparison. This is a significant underestimation. What impact does this then have on the visibility and hence the LVD calculations? The authors also need to discuss in more detail the impacts of uncertainty on the LVD and VLVD estimates. Without this level of validation, the model results cannot be used to the level of precision that the authors present in e.g. Table 2.

I would also like to see some explanation as to why the modelled visibility distance for Bangkok in Fig 4 is significantly lower than that in the observations (and in comparison to the difference at other sites), and consequently what this means for the calculation of VLVDs.

The decision that the "other pollution contribution %" is "100% minus Fire pollution contribution %" is not appropriate for the analysis that is then presented. Statements

such as those on line 336-338 and line 345-347 do not hold up. The authors need to present a justification for why the reader should assume that the model data is correct. Even so, all interpretation of non-fire LVD should probably be removed.

To aid the discussion of the changing number of LVDs further explanation of certain statements is needed. For example, Line 366-368, why is Kuching different to Singapore? Could this be because Kuching is within a fire area?

More information and explanation on the model set-up and analysis approach are needed to help the reader understand what has been done. Including (a) in section 2, further explanation about the "chemistry tracer module" is required – is there any chemistry at all? It doesn't appear so, so this is a bit misleading. It would be better to say "chemical tracer module" and be clear that the pollutants are being modelled as tracers only. The lines on p8 (163-164) describing the deposition processes could usefully be moved to this earlier point in the text. An explanation for why the domain extends so far west would also be helpful. (b) p9 line 180 – the authors need to clarify whether emissions have been injected at just 700 m or from the surface to 700 m. Is this asl or agl? (c) More detail (ideally the equations used) is needed as to how the hydroscopic growth is calculated on p11 line 232 and how this relates to the visibility calculation. Also where has the environmental relative humidity data that is used come from? This is fundamental part of the model data processing, and will introduced it's own uncertainties, but is rushed over (d) There is currently no information on how the model output has been produced for each site, so this needs to be added. For example, is it based on the modelled concentration in the lowest WRF-Chem layer for the grid box corresponding to each observation site? (e) A brief explanation as to how the runs have been conducted to identify the different source sectors is needed. Did these use labelled tracers?

The use of two different time periods for the analysis of the results for the FINN data vs the GFED data introduces differences in the outputs, which could be misinterpreted. It makes Table 3 particularly complicated to interpret. I would recommend that throughout the paper the authors only present data for the same period for all 3 model simula-
tions (i.e. 2003-2014) to avoid introducing additional uncertainty and confusion in their
results and analysis.

I would also recommend that Table 3 is modified to present the total number of days in
the 12 year period rather than an annual average, as the latter significantly distorts the
true year to year variability and introduces false precision.

The language needs some improvement particularly in the abstract and the introduc-
tion. The use of "particulate matters" rather than "matter" is somewhat unconventional.

The discussion of the role of precipitation jumps around the sections, so the authors are
encouraged to see if this could be pulled together into one, shorter overview section.
Some of the text regarding the precipitation in section 2.4 needs further explanation.
For example on line 275 more detail and/or a citation is needed for the FDDA grid
nudging. The use of mean monthly rainfall to compare the models and observations
(lines 269-274) seems strange given that the authors have nicely demonstrated the
large annual variation in rainfall timing and magnitude across the region. It would be
useful to explore whether the models are better in some seasons than others in this
region? On Line 281 the authors mention the temporal correlation, but also need to
state over what averaging period this is, e.g. is this based on daily, weekly, monthly
mean or total ppt data? Figure 3 is particularly hard to interpret. Difference plots would
be more useful here, but this figure is a candidate for removal.

Section 4 would benefit from a broader discussion of the NWP datasets, for example
there is currently no discussion of the wind fields, which are of higher order relevance
than the precipitation, particularly for the source area identification. I also find it slightly
surprising that given that the LBCs are a long way from Sumatra that WRF develops
such a discrepancy in precipitation over the central region of the domain in the differ-
ent runs. Is there a similar difference in the winds, which would therefore impact the
transport? Has any verification of the WRF wind data been conducted? This section

would benefit from being merged with the other sections on meteorology.

The attempt by the authors to use the data to assess the impact of the haze on populations in SE Asia is to be commended, but the approach taken is needlessly complicated. The units of the HED metrics are unclear and the dominance of population size on the HEDpw metric needs more careful explanation. What the results are showing are that the total number of LVDs in the region (based on observations at 50 cities) has increased over the analysis period. This conclusion could be reached without the HED and is easier to explain and understand for the reader. As explained previously the statements in this section about non-fire pollution are not justified by the approach.

The manuscript would benefit from fewer figures and I am not sure the supplementary material adds anything. The line thickness in many of the line graphs means that the bottom lines are often hidden, this is always a problem with this sort of graph, but a reduction in the line thickness would be beneficial.

Technical Corrections P2 line 45 – 99.1% is over stating the precision here. I would suggest using only 99% which is in line with the precision of other numbers given in the abstract

P4 line 66-73 – The discussion of radiative impact isn't relevant to the rest of this work, so seems unnecessary. Recommend deleting these lines.

Line 325-327 – it would be more helpful to the reader if these percentages were expressed as a number of days. The language at the end of this sentence could also be improved

Line 237 – Is the total population figure here correct? It is not clear if this the combined total, or if each city has more than 2 million?

Table 2 – The table would benefit from explanation that the VLD and VLVD for FNL_FINN and ERA_FINN are identical as they are based on observations, and that the data for FNL_GFED is different as it covers a shorter time period. However see

comments regarding making the time period consistent.

Table 2 - The FNL_FINN LVD line for Singapore does not add up to 100%.

In Table 3, the caption states that "parentheses show the fire aerosol fraction in total PM2.5" – this is very unclear and confusing. It could be taken to imply that the model also contains non-fire PM2.5, but I don't think this is the case. I think the table would be more informative and cleaner if all of the parentheses data were removed.

Figure 2 – it would be useful to highlight in the caption that all of the plots have different axes scales.

Figure 5 – the use of different axis scales in (a) is very misleading. Both data sets should be presented with the same scale and starting from 0. Where is the data that gives the green areas from? This data could usefully contribute to the discussion in the text and the validation of the model.

Figure 6 – A better way to present this data would be to have the green data as the GSOD observed LVDs and the red data as the modelled fire LVDs. This would be a more robust comparison of model vs observations and start to address issues in the comments above.

Figure 7 – the S1 and S5 line colours are too similar in my copy, so can one of these be changed please.

Figure 9 – Need to specify that these are "fire" concentrations in the caption. In this and Fig 10, the purple contours on the right hand plots prevent the underlying colours from being seen and are so small that they are unreadable, so recommend that these are removed.

Figure 11 – To ensure that there is no unintentional bias, the plot would be better if it depicted data for only 2003-2014 for all of the data sources.
* * *

---

## Author Comment (AC1) · 7 Oct 2016

**Responses to the Comments of the Anonymous Referee #1**

We very much appreciate the constructive comments and suggestions from this reviewer. Our point-by-point responses to the reviewer's comments are as follows (the reviewer's comments are marked in Italic font).

*General comments:*

*The manuscript addresses an emerging issue for Southeast Asia which concerns the impact of biomass burning on air quality and visibility. The topic is highly relevant for publication in Atmospheric Chemistry and Physics, however major issues related to the form in which the work is structured and presented (i.e. a whole rewriting of the paper is needed), clarifications in methods and analyses need to be addressed. The overall work needs to be synthesized both in the text and in the selection of the figures presented (of the 13 figures included some of them duplicate information included in other ones. If the authors want to keep all of them, they should consider moving some of the figures to the Supplementary Materials).*

Based on the reviewer's suggestion, the structure of the manuscript has been rearranged, especially in Section 2 and 3. In addition, Section 4 has been rewritten. Please note that, based on the other reviewer's suggestion, all analyses of model results and observations are now applied to the time period from 2003 to 2014.

*Specific comments:*

*Language*

*A major rewriting of the paper is needed. Several sentences are not fluent and a grammar/ punctuation check is needed. Below are some examples:*

*Line 32: remove "that"*

Done.

*Line 33: favorite should be "favourable"*

Modified to favorable.

*Line 41 and other parts: please be consistent with the tense you use. ....*

We have checked the tense throughout the manuscript.

*Line 55: "put in effect", replace with "implemented"*

Done.

*Line 82: please check your references (e.g. Miriam is the first name)*

Corrected.

*Line 118: "the great Southeast Asia" should be replaced with something similar to "over the whole Southeast Asia". Please check also elsewhere in the paper.*

Modified to "the whole Southeast Asia" throughout the manuscript.

*Line 135: please rephrase*

The sentence is revised to "Our focus in this study is on the fire aerosol life cycle. Therefore, we chose to use WRF-Chem with a modified chemical tracer module instead of a full chemistry package, to thus model the fire $PM_{2.5}$ particles as tracers without involving much more complicated gaseous and aqueous chemical processing calculations but dry and wet depositions." in Lines 118-122 of the revised version.

*Line 168: "estimations" should be always replaced with "estimates"*

Modified throughout the manuscript.

*Line 172: remove "with"*

Done.

*Line 178: "comparing" should be "compared". Please amend this everywhere in the paper.*

Modified throughout the manuscript.

*Line 190-202: please rephrase and summarize. This paragraph is too repetitive and needs to be more concise.*

The paragraph has been rephrased to "Generally speaking, there is a strong correlation between the seasonal variation of fire emissions and that of rainfall in all fire regions as shown in Fig. 2. Because mainland Southeast Asia (s1) and northern Australia (s5) are on the edge of the seasonal migration of the ITCZ, the correlation in these two regions is even more pronounced. On the other hand, in Sumatra (s2), Borneo (s3) and the rest of Maritime Continent (s4), while inter-seasonal variations of rainfall and fire emissions are still correlated with each other in general, fire emissions do exist in some raining seasons (Fig. 2b – d), owing to the precipitation features in multiple scales over these regions (e.g., the passage of MJO events) and underground peatland burning." in Lines 172-180 in the revised version.

*Line 211: units, please replace also elsewhere Line 236: "this" is missing*

Done.

*Line 294: "so that" is very often used incorrectly. Please check all the occurrences. Line 343: "are occurred", should be "occurred" Line 515: "reasons" should be "seasons".*

*Please check also other typos.*

Removed "so that" in the sentence and rephrased. Done correcting typos.

*Line 518-519: Please rephrase*

The sentence is removed. Section 4 has been rewritten in the revised version.

*Line 571-580: this section needs to be rewritten. Sentences are too long and convoluted and several grammar errors are present.*

We have rewritten Section 4.

***Methods:***

*All the introduction regarding WRF is not needed since you are using a modified version of WRF-Chem. Also you start introducing the model and have section 2.2 describing the emissions and section 2.4 discussing again the simulations. The whole method section has to be reorganized (e.g. have one section discussing the data, one on the model and one on the methods used). Please be more concise and avoid repeating the same information in different sections.*

The introduction of WRF-Chem in Section 2.1 has been condensed. We have also rearranged the structure of Section 2. Besides section 2.1, the descriptions of numerical simulations and model evaluation has been moved to Section 2.2, observation data and model derivation of visibility to Section 2.3, and the "Haze Exposure Day (HED)" definition to Section 2.4.

*Line 123: please refer more precisely to your "targeted science questions"*

The sentence has been revised to "In this study, we have used the Weather Research and Forecasting (WRF) model coupled with a chemistry component (WRF-Chem) version 3.6 (Grell et al., 2005). Our focus in this study is on the fire aerosol life cycle. Therefore, we chose to use WRF-Chem with a modified chemical tracer module instead of a full chemistry package, to thus model the fire $PM_{2.5}$ particles as tracers without involving much more complicated gaseous and aqueous chemical processing calculations but dry and wet depositions" in Lines 117-122 of the revised version.

*Line 139: you mostly focus on visibility so please also add that.*

The sentence has been revised to "This configuration lowers the computational burden substantially, and thus allows us to conduct long model integrations to determine the contributions of fire aerosol to the degradation of visibility in the region over the past decade." in Lines 123-126 of the revised version.

*Line 145: this is redundant information, please remove it.*

Removed.

*Line 146: The reported time step is for chemistry or physics?*

We have made this clearly by stating: "The time step is 180 seconds for advection and physics calculation." in Line132 of the revised version.

*Line 165: Did you only include fire emissions? Does WRF-Chem use other anthropogenic emissions?*

We only included fire $PM_{2.5}$ particles in the model; therefore, emissions of other chemical species were excluded in the simulations. To make this clearer to the reader, we have added in the manuscript that: "Therefore, we chose to use WRF-Chem with a modified chemical tracer module instead of a full chemistry package, to thus model the fire $PM_{2.5}$ particles as tracers without involving much more complicated gaseous and aqueous chemical processing calculations but dry and wet depositions." in Lines 119-122 of the revised version.

*Line 208: this should be rephrased by saying what you used for computing visibility.*

The sentence has been rephrased to "In this study, the visibility is calculated by using the *Koschmeider equation: …*" in Line 238 of the revised version.

*Line 213-216: please add a reference and rephrase*

The sentence has been modified to "Based on Eq. (1), a maximum visibility under an absolutely dry and pollution-free air is about 296 km owing to Rayleigh scattering, while a visibility in the order of 10 km is considered under a moderate to heavy air pollution by particulate matter (Visscher, 2013)." in Lines 242-245 of the revised version.

Reference:
Visscher, A. D.: Air Dispersion Modeling: Foundations and Applications, First ed., John Wiley & Sons, Inc., pp. 50, 2013.

*Line 222: please be more specific by explaining how you will use the GSOD data and to address which objectives*

We have added the explanation and also rephrased the sentence to "The observational data of visibility from the Global Surface Summary of the Day (GSOD) (Smith et al., 2011) are used in our study to identify days under particulate pollution, i.e., haze events." in Lines 250-252 of the revised version.

*Line 219: add "by increasing bext"*

The sentence has been revised to "Similarly, fire aerosols, alone or mixed with other particulate pollutants, can degrade visibility by increasing $b_{ext}$ and lead to occurrence of haze events too." in Lines 247-249 of the revised version.

*Line 225: Here you introduce model simulations, but you have a section later discussing*

*that. You should reorganize the methods and be more clear on the objectives you are addressing. "In order to compare with observations", what do you mean? Are you referring to a model evaluation? If so please explain in the relevant section how you will perform it.*

This paragraph describes the procedure of using observed visibility to evaluate modeled $PM_{2.5}$ concentrations in our study, and also the method of deriving modeled visibility based on the extinction coefficient of simulated fire aerosols as a function of particle size. We have modified the sentence to: "The observed visibility is also used to evaluate the modeled visibility and thus $PM_{2.5}$ concentration. The modeled visibility is derived based on the extinction coefficient of the fire aerosols as a function of particle size, by assuming a log-normal size distribution of accumulation mode with a standard deviation $\sigma = 2$ (Kim et al., 2008). Note that all these calculations are done for the wavelength of 550 nm unless otherwise indicated." in Lines 255-259. We have also added the details of particle hydroscopic growth calculation in Lines 264-270 of the revised version.

*Line 227: is there a reference you can quote for these assumptions? Or some local measurements used to estimate those parameters?*

We have cited Kim et al. (2008) and added this reference in the revised manuscript.

Reference:
Kim, D., Wang, C., Ekman, A. M. L., Barth, M. C., and Rasch, P. J.: Distribution and direct radiative forcing of carbonaceous and sulfate aerosols in an interactive size-resolving aerosol–climate model, Journal of Geophysical Research: Atmospheres, 113, D16309, 10.1029/2007jd009756, 2008.

*Line 225-233: this paragraph should be clarified. It is not clear how you link the discussion on fire emission composition, hygroscopic growth, etc. with your work. If it is for general overview purposes, please add it to the introduction or remove it.*

We have added more details of the visibility calculation, specifically the method to include the effect of particle hydroscopic growth in Section 2.4 of the revised version:
"To make the calculated visibility of the fire aerosols better match the reality, we have also considered hydroscopic growth of sulfate fraction of these mixed particles in the calculation based on the modeled relative humidity (*RH*). Based on Kiehl et al. (2000), the hydroscopic growth factor (*rhf*) is given by
$$rhf = 1.0 + exp\left(a_1 + \frac{a_2}{RH+a_3} + \frac{a_4}{RH+a_5}\right), \tag{2}$$
where $a_1$ to a5 are fitting coefficients given by 0.5532, -0.1034, -1.05, -1.957, 0.3406, respectively. The radius increase of wet particle ($r_{wet}$) due to hydroscopic growth will be
$$r_{wet} = r_{dry}{}^{rhf}, \tag{3}$$
where $r_{dry}$ is the radius of dry particle in micron."

*Line 238-239: again this is repetition of definitions already given. Please remove this from here and elsewhere in the manuscript.*

Removed.

*Line 268: what is the NCAR_FNL? You have not introduced that before. Please add a reference for all datasets used.*

We thank the reviewer for pointing out this typo. We have corrected "NCAR_FNL" to "NCEP_FNL".

*Line 267-272: this paragraph needs to be rewritten. Is there any difference between precipitation simulated with NCAR_FNL and FNL_FINN? Otherwise synthesise this result by comparing the simulations run with FNL and ERA. What does it mean "both results appear to be higher"? Please rephrase.*

We use TRMM observed precipitation to evaluate modeled rainfall in FNL_FINN and ERA-FINN. We have rewritten this paragraph. We have also added more discussions of the spatial and temporal correlations of monthly rainfall between model and observation in different seasons in Section 2.2 of the revised version.

*Line 301: LVDs and VLVDs have already been defined so avoid repetitions.*

Removed.

*Line 332: how can you distinguish the events caused by fires? Is it because your simulations do not include other anthropogenic emissions? Otherwise please explain how you conducted your analyses.*

We have revised the related descriptions. Firstly, we have emphasized that many LVDs could be induced by non-fire aerosols, therefore, modeled underestimate of $PM_{2.5}$ concentration and visibility degradation is expected. On the other hand, we used the VLVDs to specifically check the model performance because these events are known to be mainly induced by fire aerosols.

In Section 2.3 of the revised version, a largely revised paragraph now reads as: "As mentioned above, a visibility of 10 km is considered an indicator for a moderate to heavy particulate pollution. Hence a visibility of 10km in observation is used as the threshold for defining the "low visibility day (VLD)" in our study. We firstly derived the observed low visibility days in every year for a given city using the GSOD visibility data. Then, we derived the modeled low visibility days following the same procedure but using modeled visibility data that were only influenced by fire aerosols. Both the observed and modeled visibilities were then used to define the fraction of low visibility days that can be caused by fire aerosols alone. It is assumed that whenever fire aerosol *alone* could cause a low visibility day to occur, such a day would be attributed to fire aerosol caused LVD, regardless of whether other coexisting pollutants would have a sufficient intensity to cause low visibility or not. In addition to the LVD, we have also used a daily visibility of 7 km as the criterion to define the observed "very low visibility day (VLVD)". Such heavy haze events in the region are generally caused by severe fire aerosol pollution, thus we use their occurrence specifically to evaluate the model performance."

*Line 349-362: please rephrase to remove repetitions.*

We have modified the paragraph to: "The percentage of LVDs in Singapore has been rapidly increasing since 2012 (Fig. 6c). During the simulation period, this increase appears to be mostly from anthropogenic pollution other than fires, especially in 2012 and 2013. In monthly variation, similar to Kuala Lumpur, two peaks of fire aerosol influence appear in February-March and in September-October, respectively (Fig. 6g). In February and March, the trans-boundary transport of fire aerosols come from mainland Southeast Asia (s1), while in the summer monsoon season fire aerosols come from both Sumatra (s2) and Borneo (s3) (Fig. 7c). Except for the severe haze events in June 2013, VLVDs basically occur in September and October (i.e., 92%) due to both Sumatra and Borneo fires. In general, 34% of LVDs in Singapore are caused by fire aerosols in the FNL_FINN simulation and the rest by local and long-range transported pollutants (Table 3). Nevertheless, fire aerosol is still the major reason for the episodic severe haze conditions." in Lines 375-386 of the revised version.

**Results**

*Line 374-384: this part should be moved to the methods. You need to define earlier how you will conduct your analyses. Also using LVD in equation 3 might be more appropriate than C(i).*

We have moved this part to Section 2.4, the "Haze Exposure Day (HED)". We prefer to keep C(i) instead of LVD because LVD is defined as a day with visibility equal or lower than 10 km. However, C(i) represents the annual LVDs which means the sum of LVDs for each year.

*Line 432: here it would be also interesting to compare with the WHO limits (i.e. the limit for annual mean $PM_{2.5}$ is 10 μg m$^{-3}$).*

The sentence has been modified to "In the FNL_FINN simulation, the seasonal mean concentration of $PM_{2.5}$ within the planetary boundary layer (PBL) can exceed 20 μg m$^{-3}$ in this region (note that the air quality standard suggested by World Health Origination is 10 μg m$^{-3}$ for annual mean and 25 μg m$^{-3}$ for 24-h mean)." in Lines 430-433 of the revised version.

*Line 590: Section 4 should be rewritten. The way results are presented is too repetitive and convoluted. It would be also easier for the reader to have some clear sentences summarizing the skills of different models/emissions.*

Section 4 has been rewritten. The revisions are well marked in the version showing tracking results.

*Figures*

*Thirteen figures are really too many especially since most of them have several panels. Please select the most critical ones to summarize your findings and move the others to the supplementary material. Also some figures duplicate content shown in other, so either delete them or move to the supplements.*

The point has been well taken. We have moved Fig. 3, 10 and 13 in the original version to the supplementary and have removed Fig. 2 and 11.

*Figure 1: the number of vertical levels cannot be inferred from the figure, so please remove this part of the sentence from the caption. Also, the letters A-D are not easily readable. Please choose different colors.*

We have changed the caption to "Figure 1. Model domain used for simulations. The domain has 432 × 148 grid points with a horizontal resolution of 36 km. Five fire source regions marked in different colors and labeled as s1, s2, s3, s4 and s5, represent mainland Southeast Asia (s1), Sumatra and Java islands (s2), Borneo (s3), the rest of Maritime Continent (s4), and northern Australia (s5). A, B, C and D indicate the location of four selected cities: Bangkok (A), Kuala Lumpur (B), Singapore (C) and Kuching (D)."

We have enlarged the font size of the letters of A-D.

*Figure 2: PM$_{2.5}$ on the y-axis is not as subscript 2.5. It would be easier for the reader to have the whole name of the regions on top of each panel.*

Figure 2 has been removed.

*Figure 3: is this the yearly average of the daily means? The units can be put after "precipitation".*

The figure shows daily precipitation in 2006 only. We have added the units after "precipitation" as the reviewer suggested. This figure has been moved to the supplementary as Fig. S1.

*Figure 5: From panel (a) it is clear that the model highly underestimates observations and a scaling factor is needed. This has to be commented in the text. Could you also start both the y- axes from 0? A scatter plot might also help in quantifying the underestimation or please provide some more statistics for model evaluation.*

We have changed Fig. 5 (a) and (b) (the new Fig. 3 (a) and (b)) to let the y-axes start from 0. We have accepted the reviewer's suggestion to add a new scatter plot, Fig. 4, in the revised version to show observed visibility versus modeled visibility in FNL_FINN during known fire events. We have also added discussion of this new figure as:

"The surface observational data of PM$_{2.5}$ concentration among these four cities are only available in Singapore since 2013 from the National Environment Agency (NEA) of Singapore. We thus firstly used these data along with visibility data to evaluate model's performance for fire-cause haze events reported in Singapore during 2013-2014 (Fig. 3). Note that the observed PM$_{2.5}$ level reflects the influences of both fire and non-fire aerosols, whereas the modeled PM$_{2.5}$ only includes the impact of fire aerosols. We find that the model still predicted clearly high PM$_{2.5}$ concentrations during most of the observed haze events, especially in June 2013, and in spring and fall seasons of 2014 (highlighted green areas), though with underestimates in particle concentration of up to

30-50%, likely due to the model's exclusion of non-fire aerosols, coarse model resolution, overestimated rainfall, or errors in the emission inventory. Figure 4 shows observed visibility versus modeled visibility in FNL_FINN during the fire events shown in Fig. 3. Note that all these events have an observed visibility lower than or equal to 10 km, or can be identified as LVDs. In capturing these fire-caused haze events, the model only missed about 22% of them, or reporting a visibility larger than 10 km in 40 out of 185 observed LVDs as marked with different color in Fig. 4. When observed visibility is between 7 and 10 km, model results appear to align with observations rather well. For cases with visibility lower than 7 km, the model captured all the events (by reporting a visibility lower than 10 km, or LVD) although often overestimated the visibility range. These results imply that the VLVDs only count a very small fraction in LVDs and thus are episodic events. It is very likely that the size of concentrated fire plumes in VLVDs might be constantly smaller than the 36 km model resolution, therefore, the model results could not reach the peak values of $PM_{2.5}$ concentrations of these plumes".

*Figure 6. What do you mean with "variation"? How did you compute it? Please also report the meaning of the color coding in the caption.*

The caption has been changed to "Figure 6. (a) – (d) The percentage of LVDs per year derived using from GSOD visibility observations in Bangkok, Kuala Lumpur, Singapore, and Kuching, respectively. (e) – (h) The percentage of LVDs averaged over 2003-2014, derived using GSOD visibility observations in Bangkok, Kuala Lumpur, Singapore, and Kuching, respectively. Each bar presents the observed LVDs in each year or month. Red color shows the partition of fire-caused LVDs (captured by model) while green color presents non-fire LVDs (observed – modeled)."

*Figure 7: Please define "variation" or rephrase. Please do the same for all other figures presenting that wording.*

The caption has been changed to "Figure 7. The mean fire $PM_{2.5}$ concentrations attributed to different emission regions (s1 - s5) in: (a) Bangkok, (b) Kuala Lumpur, (c) Singapore and (d) Kuching, are all derived from FNL_FINN simulation and averaged over the period of 2003-2014."

*Figure 8: (a) please rephrase saying that the size of the circles indicates the number of days and the colors refer to specific population weights. (b) Please add units on y-axes and mention in the caption the use of different scales.*

Added units and days on y-axis. The caption has been changed to "Figure 8. (a) The mean low visibility days (circles) per year from 2003 to 2014 in 50 ASEAN cities. The size of the circles indicates the number of days. The colors refer to population-weighted fraction in the total Haze Exposure Days (HED). (b) Annual population-weighted HED ($HED_{pw}$) and arithmetic mean HED ($HED_{ar}$). Fire-caused HED are labeled as $fHED_{pw}$ and $fHED_{ar}$. Units are in days. Note that the y-axes are in different scales."

*Figure 9: region s1-s5 are not reported on the panels, so please remove them from the caption and simplify the caption as well. Also it is not clear why you report the results*

*separately by region instead of on one single figure. Figure 9 is essentially identical to Figure 10 averaging on a different period, so you can have just a four panels figure with on each panel a map showing different seasons and the 5 regions together and two panels with the same for wet scavenging. Otherwise you need to move one of the two figures to the supplements.*

We have removed s1-s5 in the caption and removed lines in (f)-(g). We actually have moved Fig. 10 to the supplementary.

*Figure 11: this is again a repetition of Figure 7. Either you condense the information in one figure or move some of the material to the supplements. It is very hard to keep in mind so many similar figures and your key message is not delivered effectively.*

The reviewer's suggestion has been well taken. We have removed the Fig. 11 in the revised version.

*Figure 12: Why do you have y-axes with negative numbers? You are displaying PM concentrations and precipitation, so your minimum value should be zero. This figure again contains information already presented (Figure 11, 7, 13), so please try and condense the figures or move them to the supplements. The captions of all figures should be also more informative on the message you want to deliver to the reader.*

We have changed all the y-axes scales to start from 0. We have also removed the original Fig. 2. This discussed figure (i.e., original Fig. 12) now becomes Fig. 2 in the revised version.

Reference:
Grell, G. A., Peckham, S. E., Schmitz, R., McKeen, S. A., Frost, G., Skamarock, W. C., and Eder, B.: Fully coupled "online" chemistry within the WRF model, Atmospheric Environment, 39, 10.1016/j.atmosenv.2005.04.027, 2005.
Kiehl, J. T., Schneider, T. L., Rasch, P. J., Barth, M. C., and Wong, J.: Radiative forcing due to sulfate aerosols from simulations with the National Center for Atmospheric Research Community Climate Model, Version 3, Journal of Geophysical Research: Atmospheres, 105, 1441-1457, 10.1029/1999JD900495, 2000.
Kim, D., Wang, C., Ekman, A. M. L., Barth, M. C., and Rasch, P. J.: Distribution and direct radiative forcing of carbonaceous and sulfate aerosols in an interactive size-resolving aerosol–climate model, Journal of Geophysical Research: Atmospheres, 113, D16309, 10.1029/2007jd009756, 2008.
Smith, A., Lott, N., and Vose, R.: The Integrated Surface Database: Recent Developments and Partnerships, Bulletin of the American Meteorological Society, 92, 704-708, doi:10.1175/2011BAMS3015.1, 2011.
Visscher, A. D.: Air Dispersion Modeling: Foundations and Applications, First ed., John Wiley & Sons, Inc., pp. 50, 2013.

---

## Author Comment (AC2) · 7 Oct 2016

**Responses to the Comments of the Anonymous Referee #2**

We very much appreciate the constructive comments and suggestions from this reviewer. The following are our point-by-point responses to the reviewer's comments (the reviewer's comments are marked in Italic font).

*General comments:*

*The paper provides a correlation of modeled particulate matter with low visibility days recorded at observation sites across South East Asia. Information is presented about the most likely source areas for biomass burning pollution for different cities and different seasons.*

*This is an interesting application of an alternative observation dataset for assessing the impact of biomass burning haze on the region and for validating CTM and dispersion models. However, the significant flaw in the way the results are presented is that the model is assumed to be correct and that all low visibility days that are not modeled are therefore specified to be due to other pollution contributions. The validity of this assumption is not demonstrated. It is quite possible that the model is over-estimating the biomass contribution at some sites and underestimating it at others. Fig 6 for example would suggest that the model may not be capturing up to 50% of the fire haze days, and Fig 4 would suggest that the model misses 50% of the VLVDs at Singapore. The references in the text to fire and non-fire LVD are therefore misleading. The authors need to reconsider how they interpret this data and present it in the paper.*

We are fully aware of the uncertainty of our model due to factors including emissions, model resolution, and meteorological fields. The uncertainty of modeling was repeatedly indicated in the manuscript, and the additional simulations using different emission inventories and meteorological fields were all designed and conducted for the purpose of identifying, at least partially, the influences of these uncertainty factors on modeled results. Nevertheless, the reviewer's point is well taken. We have made our best effort to reiterate the model uncertainty and evaluation in the revised manuscript. In addition, we have specifically indicated in many places that the model's overestimates in visibility range (underestimates in visibility degradation) are likely due to the fact that observed visibility reflects contributions of both fire and non-fire aerosols.

We have revised the description in Section 2.3 regarding our method to attribute low visibility events to fire aerosols (such events can be induced by either fire or non-fire aerosol alone or in combination), as: "As mentioned above, a visibility of 10 km is considered an indicator for a moderate to heavy particulate pollution. Hence a visibility of 10km in observation is used as the threshold for defining the "low visibility day (VLD)" in our study. We firstly derived the observed low visibility days in every year for a given city using the GSOD visibility data. Then, we derived the modeled low visibility days following the same procedure but using modeled visibility data that were only influenced by fire aerosols. Both the observed and modeled visibilities were then used to define the fraction of low visibility days that can be caused by fire aerosols alone. It is assumed that whenever fire aerosol *alone* could cause a low visibility day to occur,

such a day would be attributed to fire aerosol caused LVD, regardless of whether other coexisting pollutants would have a sufficient intensity to cause low visibility or not. In addition to the LVD, we have also used a daily visibility of 7 km as the criterion to define the observed "very low visibility day (VLVD)". Such heavy haze events in the region are generally caused by severe fire aerosol pollution, thus we use their occurrence specifically to evaluate the model performance". In addition, we have revised statements of fire aerosol contribution to contain "up to" whenever necessary.

Furthermore, the descriptions of model evaluation based on model-observation comparison have been revised, two new or largely revised paragraphs in the revised manuscript are added in Section 3.1, they provide the procedure and present the uncertainty of the model in a greater detail and clarity:

"The surface observational data of $PM_{2.5}$ concentration among these four cities are only available in Singapore since 2013 from the National Environment Agency (NEA) of Singapore. We thus firstly used these data along with visibility data to evaluate model's performance for fire-cause haze events reported in Singapore during 2013-2014 (Fig. 3). Note that the observed $PM_{2.5}$ level reflects the influences of both fire and non-fire aerosols, whereas the modeled $PM_{2.5}$ only includes the impact of fire aerosols. We find that the model still predicted clearly high $PM_{2.5}$ concentrations during most of the observed haze events, especially in June 2013, and in spring and fall seasons of 2014 (highlighted green areas), though with underestimates in particle concentration of up to 30-50%, likely due to the model's exclusion of non-fire aerosols, coarse model resolution, overestimated rainfall, and errors in the emission inventory. Figure 4 shows observed visibility versus modeled visibility in FNL_FINN during the fire events shown in Fig. 3. Note that all these events have an observed visibility lower than or equal to 10 km, and are identified as LVDs. In capturing these fire-caused haze events, the model only missed about 22% of them, or reporting a visibility larger than 10 km in 40 out of 185 observed LVDs as marked with purple color in Fig. 4. When observed visibility is between 7 and 10 km, model results appear to align with observations rather well. For cases with visibility lower than 7 km, the model captured all the events (by reporting a visibility lower than 10 km, or LVD) although often overestimated the visibility range. These results imply that the VLVDs only count a very small fraction in VLDs and thus are episodic events. It is very likely that the size of concentrated fire plumes in VLVDs might be constantly smaller than the 36 km model resolution; therefore, the model results could not reach the peak values of $PM_{2.5}$ concentrations of these plumes.

Furthermore, the LVDs in the four selected near-fire-site cities during the fire seasons from 2003 to 2014 have been identified using the daily GSOD visibility database and then compared with modeled results (Fig. 5). It is difficult to identify all the fire caused haze events beyond Singapore even in recent years. However, in Southeast Asia, severe haze events equivalent to the VLVDs in visibility degradation are known to be largely caused by fire aerosol pollution. Therefore, we used the observed VLVDs in the four selected cities to evaluate the performance of the model. We find that the modeled result displays a good performance in capturing observed VLVDs despite an overestimate in visibility range during certain events compared with the observation. The model in

general only missed about 10% or fewer VLVDs observed in the past decade (Table 3; Fig. 5). In addition, the model has reasonably captured the observed LVDs despite certain biases (Fig. 5), likely due to the fact that fire aerosol might not be the only reason responsible for the degradation of visibility during many LVDs".

*The paper would benefit from some reorganization of the sections and a reduction in the number of figures.*

Based on the reviewer's suggestion, we have reorganized the manuscript. Specifically, Section 2 and 3. Section 4 has been rewritten.

***Specific Comments:***

*Following on from the general comments, I am concerned that no real attempt at model validation is made within this paper. An additional source of observed data, e.g. PM10 concentrations, from a minimum of one of the sites (ideally many more) is needed to demonstrate that the WRF-Chem simulations are correctly capturing the fire component. The data shown in Fig 5(a) is misleading due to the use of different scales and a more robust analysis of this data is needed earlier in the paper. In fact this data may reveal useful information about missing "background" PM from the model. There are statements on line 320 that the model is underestimating PM2.5 concentration by up to 30-50% in this comparison. This is a significant underestimation. What impact does this then have on the visibility and hence the LVD calculations? The authors also need to discuss in more detail the impacts of uncertainty on the LVD and VLVD estimates. Without this level of validation, the model results cannot be used to the level of precision that the authors present in e.g. Table 2.*

We appreciate the reviewer's comments. However, as perhaps the reviewer is well aware, observational data of aerosols in Southeast Asia are still quite limited. This is also a reason why we used surface visibility data (a proxy data of $PM_{2.5}$) in the study. Besides $PM_{2.5}$ data in Singapore, there are some $PM_{10}$ monitoring data in Thailand and Malaysia. However, these are not the best data for visibility calculation due to a lack of knowledge of size distribution, not mentioning the sparseness of these data.

As reported in the paper, our model evaluation contains two parts: one is on modeled meteorological features and the other is on fire $PM_{2.5}$. Accepting the reviewer's suggestion, the detail discussion of meteorology evaluation including precipitation and wind field is now presented in Section 2.2 of the revised version.

Regarding the underestimate of $PM_{2.5}$ concentration by up to 30-50% compared to observation as shown in Fig. 5 (a) (new Fig. 3(a)), our response to the reviewer's general comments along with the newly added paragraphs in 3.1 should also address this specific comment. After all, observed $PM_{2.5}$ concentrations still reflect the contributions from other besides fire aerosols. We have added statements to indicate this fact in the revised manuscript.

We have also adjusted the scales of Fig. 5 (now Fig. 3).

*I would also like to see some explanation as to why the modeled visibility distance for Bangkok in Fig 4 is significantly lower than that in the observations (and in comparison to the difference at other sites), and consequently what this means for the calculation of VLVDs.*

Thanks for asking this interesting question. The reason why the modeled visibility in Bangkok is lower than observation in certain time period can be explained by Fig. 2 in the revised version and Fig. S5a in the supplementary section. We find that fire $PM_{2.5}$ emissions in FINNv1.5 are about a factor of 2 or 3 higher than those in GFEDv4.1s in mainland Southeast Asia (s1) during fire seasons. Note that such a difference between the two emission inventories does not show in other fire sites, i.e., s2 – s4. This implies that FINNv1.5 likely overestimated the fire emissions in mainland Southeast Asia and thus this leads to a modeled visibility in our FNL_FINN lower than observation in Bangkok. We have added the discussion in Section 4 of the revised manuscript as: "Compared to FINNv1.5, fire emissions in GFEDv4.1s over mainland Southeast Asia are more than 66% lower (Fig. 2a), and this results in a 43% lower fire $PM_{2.5}$ concentration in Bangkok (Table 4). The lower fire $PM_{2.5}$ concentration in FNL_GFED actually produces a visibility that matches better with observations in Bangkok comparing to the result of FNL_FINN (Fig. S5a). This implies that the fire emissions in FINNv1.5 are perhaps overestimated in mainland Southeast Asia".

*The decision that the "other pollution contribution %" is "100% minus Fire pollution contribution %" is not appropriate for the analysis that is then presented. Statements such as those on line 336-338 and line 345-347 do not hold up. The authors need to present a justification for why the reader should assume that the model data is correct. Even so, all interpretation of non-fire LVD should probably be removed.*

Our analysis only implies that "by considering fire aerosol alone" how many LVDs can be attributed to fire particulate pollution. We actually emphasized this point in many places of the original manuscript. The reviewer's point is well taken. To further avoid the misunderstanding, we have made it even more clearly in the revised manuscript by: (1) laying out more details about our judgment making, (2) clarifying that other cases are those that cannot be explained by fire aerosol alone, and (3) adding "up to" in the statements when necessary when referring to fire aerosol contribution. In addition, we have made our best effort to indicate that all these implications do not need to assume a perfect model to achieve.

*To aid the discussion of the changing number of LVDs further explanation of certain statements is needed. For example, Line 366-368, why is Kuching different to Singapore? Could this be because Kuching is within a fire area?*

We appreciate the reviewer's suggestion. We have stated "Kuching is in the coastal area of Borneo so Kuching is directly affected by Borneo fire events (s3)", and also "Because of its geographic location, Kuching is affected heavily by local fire events during the fire season (Fig. 7d). Fire aerosols can often degrade the visibility to below 7 km and can even reach 2 km (Fig. 3d)" in the revised version.

*More information and explanation on the model set-up and analysis approach are needed to help the reader understand what has been done. Including (a) in section 2, further explanation about the "chemistry tracer module" is required – is there any chemistry at all? It doesn't appear so, so this is a bit misleading. It would be better to say "chemical tracer module" and be clear that the pollutants are being modeled as tracers only. The lines on p8 (163-164) describing the deposition processes could usefully be moved to this earlier point in the text. An explanation for why the domain extends so far west would also be helpful. (b) p9 line 180 – the authors need to clarify whether emissions have been injected at just 700 m or from the surface to 700 m. Is this asl or agl? (c) More detail (ideally the equations used) is needed as to how the hydroscopic growth is calculated on p11 line 232 and how this relates to the visibility calculation. Also where has the environmental relative humidity data that is used come from? This is fundamental part of the model data processing, and will introduced it's own uncertainties, but is rushed over (d) There is currently no information on how the model output has been produced for each site, so this needs to be added. For example, is it based on the modeled concentration in the lowest WRF-Chem layer for the grid box corresponding to each observation site? (e) A brief explanation as to how the runs have been conducted to identify the different source sectors is needed. Did these use labeled tracers?*

(a) The sentence has been changed to "to thus model the fire $PM_{2.5}$ particles as tracers without involving much more complicated gaseous and aqueous chemical processing calculations but dry and wet depositions." We have also moved the description of deposition calculation to this place in Lines 120-122 of the revised version.

(b) We have changed the sentence to: "Therefore, we have limited the plume injection height of peat fire by a ceiling of 700 m above the ground in this study based on Tosca et al. (2011). The vertical distribution of emitted aerosols is calculated using the plume model." in Lines 160-162 of the revised version.

(c) We have added the calculation of hydroscopic growth factor and the radius increase adjustment after hydroscopic growth in Eq. (2) and (3) in the revised version. The data of relative humidity for the hydroscopic growth calculation are from the model results.

"We also consider hydroscopic growth of sulfate fraction of these mixed particles in the calculation based on the modeled relative humidity (*RH*). Based on Kiehl et al. (2000), the hydroscopic growth factor (*rhf*) is given by

$$rhf = 1.0 + exp\left(a_1 + \frac{a_2}{RH+a_3} + \frac{a_4}{RH+a_5}\right),$$ (2)

where $a_1$ to a5 are fitting coefficients given by 0.5532, -0.1034, -1.05, -1.957, 0.3406, respectively. The radius increase of wet particle (*$r_{wet}$*) due to hydroscopic growth will be

$$r_{wet} = r_{dry}{}^{rhf},$$ (3)

where $r_{dry}$ is the radius of dry particle in micron." has been added in Section 2.4 in the revised version.

(d) The fire $PM_{2.5}$ concentration presented in the paper is averaged within the PBL for the grid box corresponding to each observation site. This information has been added in the caption of Fig. 7 and 9.

(e) Yes, we labeled tracers from each source region when we created fire emission in WRF-Chem inputs. This is actually described in the emissions section, Section 2.1.

*The use of two different time periods for the analysis of the results for the FINN data vs. the GFED data introduces differences in the outputs, which could be misinterpreted. It makes Table 3 particularly complicated to interpret. I would recommend that throughout the paper the authors only present data for the same period for all 3 model simulations (i.e. 2003-2014) to avoid introducing additional uncertainty and confusion in their results and analysis.*

The reviewer's suggestion is well taken. All discussion and data in the revised manuscript are now presented from 2003 to 2014.

*I would also recommend that Table 3 is modified to present the total number of days in the 12 year period rather than an annual average, as the latter significantly distorts the true year to year variability and introduces false precision.*

We believe the reviewer's comment applies to Table 2 not Table 3 in the original version. Actually, the percentage values used in current Tables (i.e., mean LVDs/365 x 100%) serve the same purpose to describe the haze situation in any given year as suggested by the reviewer. The standard deviation shows year to year variation.

*The language needs some improvement particularly in the abstract and the introduction. The use of "particulate matters" rather than "matter" is somewhat unconventional.*

We thank the reviewer's comment and we have tried our best to polish the language of the manuscript.

*The discussion of the role of precipitation jumps around the sections, so the authors are encouraged to see if this could be pulled together into one, shorter overview section. Some of the text regarding the precipitation in section 2.4 needs further explanation. For example on line 275 more detail and/or a citation is needed for the FDDA grid nudging. The use of mean monthly rainfall to compare the models and observations (lines 269-274) seems strange given that the authors have nicely demonstrated the large annual variation in rainfall timing and magnitude across the region. It would be useful to explore whether the models are better in some seasons than others in this region? On Line 281 the authors mention the temporal correlation, but also need to state over what averaging period this is, e.g. is this based on daily, weekly, monthly mean or total ppt data? Figure 3 is particularly hard to interpret. Difference plots would be more useful here, but this figure is a candidate for removal.*

The reviewer's point is well taken. We have added the discussion about the evaluation of simulated rainfall and wind field and moved them all to Section 2.2. We have also added Table 2 in the revised version to present the spatial and temporal correlation of monthly rainfall between model and observation in different season.

The original Fig. 3 has been moved to the supplementary.

*Section 4 would benefit from a broader discussion of the NWP datasets, for example there is currently no discussion of the wind fields, which are of higher order relevance than the precipitation, particularly for the source area identification. I also find it slightly surprising that given that the LBCs are a long way from Sumatra that WRF develops such a discrepancy in precipitation over the central region of the domain in the different runs. Is there a similar difference in the winds, which would therefore impact the transport? Has any verification of the WRF wind data been conducted? This section would benefit from being merged with the other sections on meteorology.*

We have added a discussion of the surface wind difference in Section 2.2 along with related figures (Fig. S2 and S3) in the supplementary. Figure S2 and S3 show the surface wind of reanalysis data of FNL and ERA in the summer and winter monsoon seasons and the difference between FNL_FINN and ERA_FINN modeled winds. In responding to the reviewer's suggestion, we have also added discussions of the mesoscale wind pattern change in Section 2.2 besides rainfall evaluation. The discussion about the impacts of different meteorology inputs on modeled $PM_{2.5}$ concentration and LVDs are presented in Section 4 of the revised manuscript.

*The attempt by the authors to use the data to assess the impact of the haze on populations in SE Asia is to be commended, but the approach taken is needlessly complicated. The units of the HED metrics are unclear and the dominance of population size on the $HED_{pw}$ metric needs more careful explanation. What the results are showing are that the total number of LVDs in the region (based on observations at 50 cities) has increased over the analysis period. This conclusion could be reached without the HED and is easier to explain and understand for the reader. As explained previously the statements in this section about non-fire pollution are not justified by the approach.*

Haze Exposure Day (HED) can be defined by the population weighted or arithmetic mean over the included cities. The latter perhaps is the format suggested by the reviewer. As shown in the paper, we have provided results of both. The population weighted exposure is commonly used in health and policy analyses because it clearly indicates the impact correlated to population distribution. The meanings of both types of HED have been described along with their definition. The reviewer's point is well taken and we have made our best effort to clarify the implication of our results relating to fire aerosols.

*The manuscript would benefit from fewer figures and I am not sure the supplementary material adds anything. The line thickness in many of the line graphs means that the bottom lines are often hidden, this is always a problem with this sort of graph, but a reduction in the line thickness would be beneficial.*

We thank the reviewer's suggestion. We have moved the Fig. 3, 10 and 13 in the original version to the supplementary and have removed Fig. 2 and 11. All y-axes in the figures have been set to start from zero in the revised version.

***Technical Corrections***

*P2 line 45 – 99.1% is over stating the precision here. I would suggest using only 99%*

*which is in line with the precision of other numbers given in the abstract.*

Modified.

*P4 line 66-73 – The discussion of radiative impact isn't relevant to the rest of this work, so seems unnecessary. Recommend deleting these lines.*

We have shortened the discussion of radiative impact of fire aerosols in the Introduction.

*Line 325-327 – it would be more helpful to the reader if these percentages were expressed as a number of days. The language at the end of this sentence could also be improved.*

The sentence has been modified to "We find that the annual mean LVDs in Bangkok has increased from 47% (172 days per year) in the first 5-year period of the simulation (2003-2007) to 74% (272 days per year) in the last 5-year period (2010-2014). The LVDs caused by fire aerosols has increased as well (Fig. 6a)." in Lines 352-355 of the revised version.

*Line 237 – Is the total population figure here correct? It is not clear if this the combined total, or if each city has more than 2 million?*

There is no population figure presented in the paper. We are not sure to which figure the reviewer was referred. The population information of 50 ASEAN cities has been added in the supplementary (Table S1) in the revised version.

*Table 2 – The table would benefit from explanation that the VLD and VLVD for FNL_FINN and ERA_FINN are identical as they are based on observations, and that the data for FNL_GFED is different as it covers a shorter time period. However see comments regarding making the time period consistent.*

The caption of Table 3 in the revised version has been changed to "Annual mean low visibility days (LVDs; observed visibility ≤ 10 km) and very low visibility days (VLVDs; observed visibility ≤ 7 km) per year in Bangkok, Kuala Lumpur, Singapore and Kuching during 2003-2014 are presented in the second column. Parentheses show the percentage of year. The third and fourth columns show the percentage contributions along with standard deviations of fire and non-fire (other) pollutions for total low visibility days."

*Table 2 - The FNL_FINN LVD line for Singapore does not add up to 100%.*

In the revised version, the data have been changed to 36% and 64% based on the analysis from 2003 to 2014.

*In Table 3, the caption states that "parentheses show the fire aerosol fraction in total PM2.5" – this is very unclear and confusing. It could be taken to imply that the model also contains non-fire PM2.5, but I don't think this is the case. I think the table would be more informative and cleaner if all of the parentheses data were removed.*

We would like to keep the information of the percentage of fire aerosol contribution from each source region in the table. We have modified the caption to "Parentheses show the percentage of fire $PM_{2.5}$ contribution originating from each source region." to clarify the meaning in the parentheses.

*Figure 2 – it would be useful to highlight in the caption that all of the plots have different axes scales.*

Highlighted as suggested. Figure 2 has been removed to reduce the number of figures in the manuscript.

*Figure 5 – the use of different axis scales in (a) is very misleading. Both data sets should be presented with the same scale and starting from 0. Where is the data that gives the green areas from? This data could usefully contribute to the discussion in the text and the validation of the model.*

We now use the same scales starting from zero. The haze events highlighted in green are manually selected based on observed $PM_{2.5}$ concentration and visibility. A detailed discussion has been added in Section 3.1.

*Figure 6 – A better way to present this data would be to have the green data as the GSOD observed LVDs and the red data as the modeled fire LVDs. This would be a more robust comparison of model vs. observations and start to address issues in the comments above.*

We very much appreciate the reviewer's suggestion. However, since the observations actually contain both fire and non-fire contributions, therefore, we believe the current column charts present the results rather well. In this figure, each column presents the observed LVDs in each year or month. For example, in Fig. 6a of the revised version, column 2003 shows 40% observed LVDs (greed + red), which includes 10% fire LVDs (red) and 30% other LVDs (green).

*Figure 7 – the S1 and S5 line colors are too similar in my copy, so can one of these be changed please.*

Changed the s5 line color to orange.

*Figure 9 – Need to specify that these are "fire" concentrations in the caption. In this and Fig 10, the purple contours on the right hand plots prevent the underlying colors from being seen and are so small that they are unreadable, so recommend that these are removed.*

We have modified the caption to contain "fire $PM_{2.5}$ concentration". We have also removed the contour lines in Fig. 9 (f) – (g) and Fig. S4 (f) – (g).

*Figure 11 – To ensure that there is no unintentional bias, the plot would be better if it depicted data for only 2003-2014 for all of the data sources.*

We have removed this figure in the revised version.

---

## Referee Report (RR1)

**Review of manuscript doi:10.5194/acp-2016-504, 2016**

**Biomass Burning Aerosols and the Low Visibility Events in Southeast Asia**

**by**

**H. Lee, R.Z. Bar-Or and C. Wang**

**General comments:**

The authors have put a lot of effort to improve the manuscript that is now clearer in terms of both presentation and analyses. However I recommend major changes to be made to consider the manuscript suitable for publication. Please refer to the detailed comments below to improve presentation of the results. Some restructuring of the manuscript is still needed especially in the methods/results sections.

**Specific comments:**

Line 62: A paper on health impacts from fires was recently published in Scientific Reports that could be worth citing (www.nature.com/articles/srep37074):

"Population exposure to hazardous air quality due to the 2015 fires in Equatorial Asia"

by P. Crippa, S. Castruccio, S. Archer-Nicholls, G. B. Lebron, M. Kuwata, A. Thota, S. Sumin, E. Butt, C. Wiedinmyer & D. V. Spracklen

Line 80: please rephrase "various climate variabilities…in different temporal scales"

Line 81: ENSO is actually "El Niño–Southern Oscillation" (remove "and" and type Niño correctly)

Line 133: replace "included" with "adopted"

Line 177-180: please rephrase the last part of the sentence

Line 181: The first part of this section still refers to model settings so should be merged with section 2.1. From Line 198 the authors are discussing model evaluation with respect to precipitation, so this discussion should be moved either at the beginning of the results or merged to section 4 when the influence of different meteorological boundary conditions is anlayzed. Please also consider summarizing it.

Line 324: "highlighted green areas" could be removed unless you explicitly refer to Figure 3.

Line 408: change "in causing degradation of air quality" with "in degrading air quality"

Line 413: "haze event occurrence across from…" this part is not clear, please rephrase

Line 419-422: You are not accounting for other anthropogenic emissions in your simulations, so this sentence should be supported by a literature reference and possibly linked to any evidence/results in your paper.

Line 436: is this distance inferred from Figure 9? If so please refer to the figure or provide appropriate reference.

Line 482: wrong figure number. It should be Fig 7b and c.

Line 493: replace "similar to" with "similarly to", also in other parts of the manuscript

Line 494: It is not clear how you are able to infer the contribution from different regions. You should mention this somewhere in the methods. In general all the presented results should be supported by a clear explanation on how have been derived in an appropriate method section.

Line 496: Section 3.3 is too long and dispersive. The authors present the role of winds, wet scavenging, quantify the contribution of fire emissions to different regions/cities and finally introduce the role of different emission inventories. Please consider reorganizing and summarizing. The paragraph from line 496 could be moved to the model description or merged to the discussion on the emission inventories. You could separate these sections and have one for meteorological and one for emissions influence.

Line 510: "wind field" should be "wind fields", check this through the paper.

Line 514-523: this part could be integrated with the content in section 2.2 where the authors evaluate the model in terms of precipitation. However I recommend organizing a new section where the model evaluation is discussed.

Line 532: which "modelled results"? please be more specific/rephrase, also with respect to the subsequent sentence starting with "To examine such an influence".

Line 548-549: please rephrase and be more specific on which modeled results you are referring to.

Line 588: It would be better to summarize your findings instead of mentioning what you have done.

From line 591: please check the use of tenses.

Line 599: it would be good to add how many of these events were likely to be due to fires.

Line 601: rephrase as "but also in those". "Pollutions" should be "pollution"

Line 602: please rephrase this sentence and link better to the previous conclusions.

Line 606: remove "as well"

**Tables:**

Table 2: Please provide more accurate description in the caption. Refer to the TRMM dataset used for observations and mention the different model runs.

Table 3: Please mention that the table includes comparison of different model runs/emissions. Also there are several typos (e.g. "VLD" instead of "LVD"). More details on how the terms in the last column are computed should be provided either in the caption or in the methods. If it is simply the difference from 100% maybe the whole column could be removed. There is an error in the last column FNL_FINN for Bangkok since fire and other pollution contribution appear to be the same.

Table 4. Please rephrase the caption. "Annual mean and standard deviation contributed by each source" is not very clear. Line 813 needs to be changed with "Regions s1-s5 are defined in Fig 1.

**Figures**

Figure 2: This figure can be still improved. The labels on the x axis should be more frequent and regularly spaced, otherwise it is impossible to infer the months/years of any episode of interest. You could have labels such as mm/yy and regular ticks on the x axis (at least every 6 months or every year). It would be also better to have the same y axis at least for precipitation (0-25) for easier comparison. Although the y-axis for $PM_{2.5}$ cannot be the same for all regions, I am wondering if it is possible to have at least panel b-e on the same scale (0-15) for better comparison and leave panel a up to 40 and mention this in the caption. It would be also easier for the reader to have direct reference the region associated with each panel by adding s1,s2…on top of each panel.

Figure 3: rephrase b by simplifying the sentence (e.g. visibility from GSO observations…and FNL_FINN simulations…)

Figure 4: Panel 4b is missing, so not sure what line 851 refers to. From "Data points marked with purple" please rephrase. How are those known fire events identified?

Figure 7: this figure can be improved by placing a frame around each panel and grid lines every month.

Figure 9: Panels h to j are not described in the caption. Please add details.

**Supplementary Materials:**

Figure S2: ERA-Interim is spelled wrong (also in Fig S3)

Figure S5: you should define "A-S-O-F-M-A" in the caption or refer better to the fire seasons.

Figure S6: panels a and b should be improved by placing a frame around each panel and grid lines regularly spaced. Since you are focusing on June-July 2013 the labels could include just the day and month.

---

## Author Response (AR2)

**Responses to the Comments of the Anonymous Referee #1**

We appreciate the further useful comments and suggestions from this reviewer again. Our point-by-point responses to the reviewer's comments are as follows (the reviewer's comments are marked in Italic font). The tracking version of the manuscript is attached with this reply.

*General comments:*

*The authors have put a lot of effort to improve the manuscript that is now clearer in terms of both presentation and analyses. However I recommend major changes to be made to consider the manuscript suitable for publication. Please refer to the detailed comments below to improve presentation of the results. Some restructuring of the manuscript is still needed especially in the methods/results sections.*

Based on the reviewer's suggestion, the structure of the manuscript has been rearranged, especially in Section 2 and 4. Section 2.2 in the original manuscript has been removed and the contents have been merged into Section 2.1 and 4.1, respectively. In the revised version, Section 4.1 and 4.2 focus on the influence of different meteorological input datasets and different fire emission inventories on fire aerosol abundance separately.

**Specific comments:**

*Line 62: A paper on health impacts from fires was recently published in Scientific Reports that could be worth citing (www.nature.com/articles/srep37074): "Population exposure to hazardous air quality due to the 2015 fires in Equatorial Asia" by P. Crippa, S. Castruccio, S. Archer-Nicholls, G. B. Lebron, M. Kuwata, A. Thota, S. Sumin, E. Butt, C. Wiedinmyer & D. V. Spracklen*

Added.

*Line 80: please rephrase "various climate variabilities...in different temporal scales"*

The sentence has been modified to "Reid et al. (2012) investigated relationships between fire hotspot appearance and various weather phenomena as well as climate variabilities in different time scales over the MC, …"

*Line 81: ENSO is actually "El Niño–Southern Oscillation" (remove "and" and type Niño correctly)*

Corrected.

*Line 133: replace "included" with "adopted"*

Done.

*Line 177-180: please rephrase the last part of the sentence*

The sentence has been modified to "On the other hand, Sumatra (s2), Borneo (s3) and the rest of the Maritime Continent (s4) do not have clearly identifiable dry seasons and this contributes to the weaker correlation (Fig. 2b – d). Besides that, underground peatland burning may not be immediately extinguished by precipitation."

*Line 181: The first part of this section still refers to model settings so should be merged with section 2.1. From Line 198 the authors are discussing model evaluation with respect to precipitation, so this discussion should be moved either at the beginning of the results or merged to section 4 when the influence of different meteorological boundary conditions is anlayzed. Please also consider summarizing it.*

The first part of this section that discusses the design of numerical simulations has been merged to Section 2.1 "The model". The model evaluation with respect to precipitation has been moved to Section 4.1.

*Line 324: "highlighted green areas" could be removed unless you explicitly refer to Figure 3.*

Removed.

*Line 408: change "in causing degradation of air quality" with "in degrading air quality"*

Done.

*Line 413: "haze event occurrence across from..." this part is not clear, please rephrase*

The sentence has been changed to "Interestingly, the discrepancy of these two variables, however, has become smaller in recent years and even reversed in 2014, implying an increase of haze occurrence across cities with different populations in the region."

*Line 419-422: You are not accounting for other anthropogenic emissions in your simulations, so this sentence should be supported by a literature reference and possibly linked to any evidence/results in your paper.*

We added the IEA (2015) report to support our statement.

IEA: Energy and Climate Change, World Energy Outlook Special Report, International Energy Agency, pp. 74 -77, 2015.

*Line 436: is this distance inferred from Figure 9? If so please refer to the figure or provide appropriate reference.*

The sentence has been modified to "Fire aerosol plumes with concentrations higher than 0.1 μg m$^{-3}$ can be transported westward as far as 7000 km from the burning sites (Fig. 9a)."

*Line 482: wrong figure number. It should be Fig 7b and c.*

Corrected.

*Line 493: replace "similar to" with "similarly to", also in other parts of the manuscript*

Corrected.

*Line 494: It is not clear how you are able to infer the contribution from different regions. You should mention this somewhere in the methods. In general all the presented results should be supported by a clear explanation on how have been derived in an appropriate method section.*

This was actually clearly described in the text in Section 2.1, as "In order to distinguish the spatial-temporal coverage and influence of biomass burning aerosols from different regions in Southeast Asia and nearby northern Australia, we have created five tracers to represent fire aerosols respectively from mainland Southeast Asia (s1), Sumatra and Java islands (s2), Borneo (s3), the rest of the Maritime Continent (s4), and northern Australia (s5) as illustrated in Fig. 1".

We now have added one more sentence to avoid any potential misunderstanding: "Based on this design, we are able to identify fire $PM_{2.5}$ concentration from different regions and estimate the contribution to the total fire $PM_{2.5}$ in a receptor city."

*Line 496: Section 3.3 is too long and dispersive. The authors present the role of winds, wet scavenging, quantify the contribution of fire emissions to different regions/cities and finally introduce the role of different emission inventories. Please consider reorganizing and summarizing. The paragraph from line 496 could be moved to the model description or merged to the discussion on the emission inventories. You could separate these sections and have one for meteorological and one for emissions influence.*

The paragraph from Line 496 has been moved to Section 4.2 with the discussion of the impact of different fire emission inventories on fire aerosol concentration.

*Line 510: "wind field" should be "wind fields", check this through the paper.*

Modified.

*Line 514-523: this part could be integrated with the content in section 2.2 where the authors evaluate the model in terms of precipitation. However I recommend organizing a new section where the model evaluation is discussed.*

We separated the discussion of meteorological datasets and fire emission inventories into Section 4.1 and 4.2.

*Line 532: which "modelled results"? please be more specific/rephrase, also with respect to the subsequent sentence starting with "To examine such an influence".*

The sentence has been modified to "In addition to meteorological inputs, using different fire emission estimates could also affect the modeled $PM_{2.5}$ concentration. To examine this impact, we have compared two simulations with the same meteorological input but different fire emission inventories, the FNL_FINN using FINNv1.5 and FNL_GFED using GFEDv4.1s."

*Line 548-549: please rephrase and be more specific on which modeled results you are referring to.*

The sentence has been modified to "We would also like to point out the importance of spatiotemporal distribution of fire emission to the modeled $PM_{2.5}$ concentration."

*Line 588: It would be better to summarize your findings instead of mentioning what you have done.*

The sentence has been changed to "Based on these results, we suggest further research is needed to improve the current estimate of the spatiotemporal distribution of fire emissions, in addition to total emitted quantities from the fire hotspots"

*From line 591: please check the use of tenses.*

Checked.

*Line 599: it would be good to add how many of these events were likely to be due to fires.*

The sentence has been changed to "The top four cities in the HED ranking, Jakarta, Bangkok, Hanoi, and Yangon, with a total population exceeding two millions, all have experienced more than 200 days per year of low visibility due to particulate pollution over the past decade and more than 50% of those low visibility days were mainly due to fire aerosols."

*Line 601: rephrase as "but also in those". "Pollutions" should be "pollution"*

Done.

*Line 602: please rephrase this sentence and link better to the previous conclusions.*

The sentence has been modified to "In summary, the fire aerosols are found to be responsible for up to about half of the total exposures to low visibility in the region."

*Line 606: remove "as well"*

Removed.

**Tables:**

*Table 2: Please provide more accurate description in the caption. Refer to the TRMM dataset used for observations and mention the different model runs.*

This table has been moved to Table 4 and the caption has been modified to "Table 4. The spatial and temporal correlation of monthly rainfall between models (FNL_FINN and

ERA_FINN) and observation (TRMM) during 2003-2014. FMA, MJJ, ASO, NDJ and All represents February-April, May-July, August-October, November-January and whole year, respectively."

*Table 3: Please mention that the table includes comparison of different model runs/emissions. Also there are several typos (e.g. "VLD" instead of "LVD"). More details on how the terms in the last column are computed should be provided either in the caption or in the methods. If it is simply the difference from 100% maybe the whole column could be removed. There is an error in the last column FNL_FINN for Bangkok since fire and other pollution contribution appear to be the same.*

Typos have been corrected in the revised manuscript. We would like to keep "other pollution contribution" in the table although it is simply the difference from 100%.

Table 2. Annual mean low visibility days (LVDs; observed visibility $\leq$ 10 km) and very low visibility days (VLVDs; observed visibility $\leq$ 7 km) per year in Bangkok, Kuala Lumpur, Singapore and Kuching during 2003-2014 are presented in the second column. Parentheses show the percentage of year. The third column shows the percentages, along with standard deviations, of low visibility days explained by fire aerosols alone (i.e. the LVDs captured by the model). The fourth column is the same as the third column but for non-fire (other) pollutions, which is calculated as 100% - fire pollution contribution (i.e. the percentage of LVDs not captured by the model)."

*Table 4. Please rephrase the caption. "Annual mean and standard deviation contributed by each source" is not very clear. Line 813 needs to be changed with "Regions s1-s5 are defined in Fig 1.*

This table has been moved to Table 3 in the revised manuscript. The caption has been modified to "Table 3. Annual mean and standard deviation of modeled fire $PM_{2.5}$ concentration ($\mu g \ m^{-3}$) in Bangkok, Kuala Lumpur, Singapore, and Kuching during 2003-2014 contributed by each source region (s1 – s5). Parentheses show the percentage of fire $PM_{2.5}$ contribution originating from each source region. Regions s1-s5 are defined in Fig. 1. FNL_FINN, ERA_FINN and FNL_GFED are three model simulations descried in Section 2.1."

**Figures**

*Figure 2: This figure can be still improved. The labels on the x axis should be more frequent and regularly spaced, otherwise it is impossible to infer the months/years of any episode of interest. You could have labels such as mm/yy and regular ticks on the x axis (at least every 6 months or every year). It would be also better to have the same y axis at least for precipitation (0-25) for easier comparison. Although the y-axis for $PM_{2.5}$ cannot be the same for all regions, I am wondering if it is possible to have at least panel b-e on the same scale (0-15) for better comparison and leave panel a up to 40 and mention this in the caption. It would be also easier for the reader to have direct reference the region associated with each panel by adding s1,s2...on top of each panel.*

The figure has been modified in the revised manuscript.

*Figure 3: rephrase b by simplifying the sentence (e.g. visibility from GSO observations...and FNL_FINN simulations...)*

The caption has been modified to "Figure 3. (a) Time series of daily surface $PM_{2.5}$ from the ground-based observations (black line) and FNL_FINN simulated results (red line) in Singapore during 2013-2014. (b) Same as (a) but daily visibility from GSOD observations (black line) and calculated result from FNL_FINN (red line). Highlighted green areas are known haze events caused by fire aerosols, which are reported by news or manually selected based on observed $PM_{2.5}$. Two gray lines mark the visibility of 7 and 10 km, respectively."

*Figure 4: Panel 4b is missing, so not sure what line 851 refers to. From "Data points marked with purple" please rephrase. How are those known fire events identified?*

4b has been corrected to 3b. The caption has been modified to "Purple points remark the known low visibility events that model failed to produce a visibility at least qualified for LVD." Those known fire events has been described in the caption of Figure 3.

*Figure 7: this figure can be improved by placing a frame around each panel and grid lines every month.*

The figure has been modified in the revised manuscript.

*Figure 9: Panels h to j are not described in the caption. Please add details.*

The caption has been modified to "Figure 9. Seasonal mean fire $PM_{2.5}$ concentration (μg $m^{-3}$) and wind within the PBL modeled in FNL_FINN during February to April, 2003–2014 for fire $PM_{2.5}$ source region from (a) mainland Southeast Asia, (b) Sumatra and Java islands, (c) Borneo, (d) the rest of the Maritime Continent, and (e) northern Australia. (f)-(j) Same as (a)-(e) but for seasonal mean wet scavenging time (days)."

**Supplementary Materials:**

*Figure S2: ERA-Interim is spelled wrong (also in Fig S3)*

Corrected.

*Figure S5: you should define "A-S-O-F-M-A" in the caption or refer better to the fire seasons.*

"F, M and A in the x-axis of (a) indicates February, March and April, respectively. A, S and O in the x-axis of (b) – (d) represents August, September, and October, respectively." has added in the caption of Figure S5 as well as Figure 5.

*Figure S6: panels a and b should be improved by placing a frame around each panel and grid lines regularly spaced. Since you are focusing on June-July 2013 the labels could*

*include just the day and month.*

The figure has been modified in the revised manuscript.

**Responses to the Comments of the Anonymous Referee #2**

We appreciate the additional comments from this reviewer. The following are our point-by-point responses to the reviewer's comments (marked in Italic font). The tracking version of the manuscript is attached with this reply.

*The manuscript has been substantially revised. Suggestions for improving the wording in a few places include:*
*Line 122: add "including" so it becomes "but including dry and wet deposition"*
Added.

*Line 172 and 177: do the authors mean "anti-correlation" here rather than correlation?*
Yes, the word has been corrected.

*Line 229: modify to "In general, the model has"*
Done.

*Line 231: change to "terrain effects" rather than effect*
Suggested change has been made.

*Line 309: modify to "We first focus"*
Done.

*Line 312-313: change to "receptor city for the fire events in mainland Southeast Asia"*
The change has been made.

*Line 318: add "the" so it becomes "to evaluate the model's"*
Done.

*Line 330: delete "or"*
Deleted.

*Line 561: change to "fire aerosols over the past decade"*
The change has been made.

*Line 603: should "exposes" be "exposures" or an equivalent?*
Changed to exposures.

*Line 604: remove "at"*
Done.

[revised manuscript text omitted]